# Doubly Robust Conditional VAE via Decoder Calibration: An Implicit KL Annealing Approach

**Chuanhui Liu** *liu2306@purdue.edu*
*Department of Statistics, Purdue University*

**Xiao Wang** *wangxiao@purdue.edu*
*Department of Statistics, Purdue University*

**Reviewed on OpenReview:** *https://openreview.net/forum?id=VIkycTWDWo*

## Abstract

Several variants of Variational Autoencoders have been developed to address inherent limitations. Specifically, $\sigma$-VAE utilizes a scaled identity matrix $\sigma^2 I$ in the decoder variance, while $\beta$-VAE introduces a hyperparameter $\beta$ to reweight the negative ELBO loss. However, a unified theoretical and practical understanding of model optimality remains unclear. For example, existing learning theories on the global optimality of VAE provide limited insight into their empirical success. Previous work showed the mathematical equivalence between the variance scalar $\sigma^2$ and the hyperparameter $\beta$ in shaping the loss landscape. While $\beta$-annealing is widely used, how to implement $\sigma$-annealing is still unclear. This paper presents a comprehensive analysis of $\sigma$-CVAE, highlighting its enhanced expressiveness in parameterizing conditional densities while addressing the associated estimation challenges arising from suboptimal variational inference. In particular, we propose Calibrated Robust $\sigma$-CVAE, a doubly robust algorithm that facilitates accurate estimation of $\sigma$ while effectively preventing the posterior collapse of $\phi$. Our approach, leveraging functional neural decomposition and KL annealing techniques, provides a unified framework to understand both $\sigma$-VAE and $\beta$-VAE regarding parameter optimality and training dynamics. Experimental results on synthetic and real-world datasets demonstrate the superior performance of our method across various conditional density estimation tasks, highlighting its significance for accurate and reliable probabilistic modeling. The implementation is publicly available at https://github.com/chuanhuiliu/calibrated_cvae.

## 1 Introduction

Conditional distributions play an essential role in characterizing the complex dependence of the data $\boldsymbol{y} \in \mathbb{R}^q$ on given labels $\boldsymbol{x} \in \mathbb{R}^p$. Canonical methods, such as regression or density estimators, face challenges when the data-generating distribution $p_{gt}(\boldsymbol{y}|\boldsymbol{x})$ is high-dimensional. Deep latent generative models based on amortized variational inference are widely used as a scalable approach to model complex distributions and scale to large datasets. Specifically, the Conditional VAE (CVAE) was introduced for conditional learning (Sohn et al., 2015), based on Variational Autoencoders (VAE) (Kingma & Welling, 2014; Rezende et al., 2014).

To model the conditional distribution $p_{gt}(\boldsymbol{y}|\boldsymbol{x})$, Gaussian $\sigma$-CVAE incorporates a Gaussian decoder $p_{\theta,\sigma}(\boldsymbol{y}|\boldsymbol{x},\boldsymbol{z}) = N(\mu_\theta(\boldsymbol{x},\boldsymbol{z}), \sigma^2 I_q)$, where $\theta$ parameterizes the decoder mean $\mu_\theta(\boldsymbol{x},\boldsymbol{z})$, and $\sigma^2$ parameterizes the decoder variance (Kingma et al., 2016; Dai & Wipf, 2018). In detail, the latent variable $\boldsymbol{z} \in \mathbb{R}^d$ is sampled from a data-independent prior $p(\boldsymbol{z}|\boldsymbol{x}) = N(0, I_d)$ (Doersch, 2021). Thus, the parametric form of $p_{\theta,\sigma}(\boldsymbol{y}|\boldsymbol{x})$ is

$$p_{\theta,\sigma}(\boldsymbol{y}|\boldsymbol{x}) = \int N(\boldsymbol{y}|\mu_\theta(\boldsymbol{x},\boldsymbol{z}), \sigma^2 I_q) N(\boldsymbol{z}|0, I_d) d\boldsymbol{z}. \tag{1}$$

Gaussian $\sigma$-CVAE also includes a Gaussian encoder, $q_\phi(\boldsymbol{z}|\boldsymbol{y},\boldsymbol{x}) = N(\mu_\phi(\boldsymbol{y},\boldsymbol{x}), \Sigma_\phi(\boldsymbol{y},\boldsymbol{x}))$, such that the log-likelihood $p_{\theta,\sigma}(\boldsymbol{y}|\boldsymbol{x})$ is replaced by a tractable Evidence Lower Bound (ELBO), i.e.,

$$\text{ELBO}(\theta,\sigma,\phi) = \mathbb{E}_{q_\phi}[\log p_{\theta,\sigma}(\boldsymbol{y}|\boldsymbol{x},\boldsymbol{z})] - \text{KL}[q_\phi(\boldsymbol{z}|\boldsymbol{y},\boldsymbol{x})||p(\boldsymbol{z})]. \tag{2}$$

With both the decoder and the encoder being Gaussian distributions, the sampling procedure of $q_\phi(\boldsymbol{z}|\boldsymbol{y},\boldsymbol{x})$ and the computation of ELBO are simplified, allowing the model to be scalable to large datasets via the reparameterization trick (Kingma & Welling, 2014; Rezende et al., 2014). In particular, the label $\boldsymbol{x}$ influences the mean of the decoder $\mu_\theta(\boldsymbol{x},\boldsymbol{z})$, assuming a data-independent prior $p(\boldsymbol{z})$ without loss of generality (Zheng et al., 2022).

The objective of $\sigma$-CVAE is to minimize the expected negative ELBO on the model parameters $\{\theta,\sigma,\phi\}$ with respect to the (empirical) data distribution $D$. Ideally, the expected negative ELBO should reach the conditional data entropy $H(Y|X)$, representing the theoretical lower bound for the model's performance.

$$\mathbb{E}_D[-\text{ELBO}(\theta,\sigma,\phi)] \geq \mathbb{E}_D[-\log p_{\theta,\sigma}(\boldsymbol{y}|\boldsymbol{x})] \geq \mathbb{E}_D[-\log p_{gt}(\boldsymbol{y}|\boldsymbol{x})] := H(Y|X) \tag{3}$$

The inequalities in Eq.3 provide the conditions under which $\theta,\sigma,\phi$ are optimal in the separate objectives of $\sigma$-CVAE—generative modeling and variational inference. The first gap in Eq.3 is referred to as *inference gap* (Cremer et al., 2018), which is defined as $\mathbb{E}_D\text{KL}[q_\phi(\boldsymbol{z}|\boldsymbol{x},\boldsymbol{y})||p_{\theta,\sigma}(\boldsymbol{z}|\boldsymbol{x},\boldsymbol{y})]$. It determines the optimality of $\phi$ given any $\theta,\sigma$ in terms of the quality of variational inference of $p_{\theta,\sigma}(\boldsymbol{z}|\boldsymbol{x},\boldsymbol{y})$. The second inequality defines the *approximation gap*, which equals $\mathbb{E}_D[\log p_{gt}(\boldsymbol{y}|\boldsymbol{x}) - \log p_{\theta,\sigma}(\boldsymbol{y}|\boldsymbol{x})] \approx \mathbb{E}_{\boldsymbol{x}}\text{KL}[p_{gt}(\boldsymbol{y}|\boldsymbol{x})||p_{\theta,\sigma}(\boldsymbol{y}|\boldsymbol{x})]$. Notably, the optimality of $\theta,\sigma$ determines how closely the generative model $p_{\theta,\sigma}(\boldsymbol{y}|\boldsymbol{x})$ of $\sigma$-CVAE approximates the data-generating density $p_{gt}(\boldsymbol{y}|\boldsymbol{x})$ in probabilistic modeling.

Given the same parameter $\theta,\phi$, the negative $\text{ELBO}(\theta,\sigma,\phi)$ loss is connected to $\beta$-VAE (Higgins et al., 2016). To be specific, it is proportional to the objective function of $\beta$-CVAE, assuming a fixed unit variance (Lucas et al., 2019b; Rybkin et al., 2021), i.e.,

$$\mathbb{E}_{q_\phi}[||\mu_\theta(\boldsymbol{x},\boldsymbol{z}) - \boldsymbol{y}||^2]/2\sigma^2 + \text{KL}[q_\phi||p(\boldsymbol{z})] \propto \mathbb{E}_{q_\phi}[||\mu_\theta(\boldsymbol{x},\boldsymbol{z}) - \boldsymbol{y}||^2] + \beta\text{KL}[q_\phi||p(\boldsymbol{z})]. \tag{4}$$

Therefore, the optimal $\beta$ is believed to be the maximum likelihood estimate of $\sigma$ (Lucas et al., 2019a; Rybkin et al., 2021). Note that the parameter $\sigma^2$ and the hyperparameter $\beta$ are distinct and not mutually exclusive.

This paper presents a comprehensive analysis of $\sigma$-CVAE, with particular attention given to $\sigma$. The organization and contributions of this paper are as follows.

(1) **A less constrained approximation theory without assumptions for optimal variational inference**. In Section 3, we analyze the approximation gap of $\sigma$-CVAE for a non-fixed $\sigma$ and establish a novel approximation theory in Theorem 3.4, leveraging Gaussian noise outsourcing (Agrawal & Domke, 2021) and block neural decomposition (Sobol, 2001). Our characterization of oracle decoders identifies and relaxes restrictive assumptions on optimal parameters $\theta,\sigma$, including $\sigma \to 0$ (Dai & Wipf, 2018; Zheng et al., 2022) or $\theta$ constrained to the linear VAE regime (Lucas et al., 2019b; Wang & Ziyin, 2022; Dang et al., 2023). Relaxing these assumptions, however, may result in the absence of an oracle encoder within the amortized Gaussian family, emphasizing how suboptimal variational inference can be underestimated in existing learning theories.

(2) **The duality between $\phi$ and $\sigma$ leads to a unified annealing**. In Section 4, we analyze the dual relationship between the parameter $\sigma$ and $\phi$ to enhance the optimization outcomes. In Section 4.1, we highlight the importance of $\phi$ in the training dynamics of $\sigma$, where $\phi$ controls the bias in the gradient(optimum) of the ELBO w.r.t. $\sigma$. On the other hand, in Section 4.2, leveraging the loss equivalence of $\beta$ in $\beta$-VAE Higgins et al. (2016), we demonstrate $\sigma$ provides a smoothing effect in preventing the posterior collapse of $\phi$. In particular, we show that KL ($\beta$)-annealing (Bowman et al., 2016; Fu et al., 2019) is essentially a sequence of fixed calibrations on the decoder variance $\sigma$. Leveraging this dual relationship between $\phi$ and $\sigma$, we argue that an accurate estimation of $\sigma$ requires a better $q_\phi$, and an explicit KL annealing scheme could be redundant. In Section 4.3, we motivate and propose Calibrated Robust $\sigma$-CVAE, a simple $\sigma$-calibration algorithm.

(3) **Calibrated Robust $\sigma$-CVAE is doubly robust.** In Section 5.1, we empirically validate that suboptimal variational inference can be the main source of numerical instability in the estimation of $\sigma$. More

importantly, we confirm the **double robustness** of our algorithm, showing that it not only provides **accurate estimates** of the variance parameter $\sigma$, but also **prevents posterior collapse** of $q_\phi$. In other words, it can efficiently explore the loss landscape of $\theta, \phi$ to prevent posterior collapse and provide robust variance estimation. Compared to existing KL annealing methods, we validate the superiority and effectiveness of $\sigma$ calibration. Starting from Section 5.2, we compared the performance of Calibrated Robust $\sigma$-CVAE in various conditional density estimation tasks, showing its superiority over other conditional learning methods.

## 2 Related Works

**Expressive decoder and approximation gap.** We consider Gaussian decoders extending from $\sigma$-VAE, where the variance is a scaled identity matrix $\Sigma = \sigma^2 I_d$. Although many recent applications assume a fixed variance $\sigma^2 = 1$ (Castrejon et al., 2019; Yamasaki et al., 2023; Loizillon et al., 2024), critics have highlighted that this assumption leads to limited generative power and a positive approximation gap. If $\sigma$ is not fixed, one line of research establishes the tightness of the approximation gap in the asymptotic regime $\sigma \to 0$ (Dai & Wipf, 2018; Zheng et al., 2022). The assumptions regarding $\sigma \to 0$ are problematic. Mattei & Frellsen (2018) shows that this can lead to an unbounded likelihood and demonstrates a zero approximation gap between Gaussian VAE and a sub-model of nonparametric mixture models (Lindsay, 1995). However, their results do not extend to conditional variants. Another independent line of research shows that the generative model under linear VAE assumptions corresponds to probabilistic Principal Component Analysis (pPCA) models (Lucas et al., 2019b; Wang & Ziyin, 2022), which also demonstrates a zero approximation gap. Linear VAE theories are rigorous but do not fully account for the success of VAE. Compared to existing learning theories of the global optimality of VAE models, our approximation theory focuses on analyzing the optimality of decoders in recovering the data-generating distribution without the above-mentioned assumptions.

**Robust decoder variance estimation.** Flexible covariance parametrization in the Gaussian decoder faces several practical challenges. The common challenge is the likelihood blow-up problem, where the maximum likelihood estimation is ill-defined as the likelihood function $\log p_{\theta,\sigma}(\boldsymbol{y}|\boldsymbol{x})$ becomes infinite or unbounded when the covariance approaches zero. For instance, Mattei & Frellsen (2018) showed that the multilayer perceptrons (MLP) parameterization of the decoder (Kingma & Welling, 2014) results in unbounded likelihood functions for continuous responses. To mitigate unbounded likelihood, Takahashi et al. (2018); Skafte et al. (2019); Stirn & Knowles (2020) assumes a conjugate gamma prior distribution for the inverse variance $1/\sigma^2$. It transforms the decoder $p_\theta(\boldsymbol{y}|\boldsymbol{x}, \boldsymbol{z})$ into a Student-t distribution. Another challenge is model non-identifiability, where different parameters of the model can yield the same likelihood or ELBO, leading to multiple (local) optima (Khemakhem et al., 2020). In addition, a dual-step optimization (Detlefsen et al., 2019; Rybkin et al., 2021) is preferred over a joint optimization of $\theta, \phi$ and the covariance of the decoder (Dai & Wipf, 2018; Takahashi et al., 2018) to improve numerical stability. We follow the standard setup of the Gaussian decoder variance, using a scaled identity matrix, and follow joint optimization (Rybkin et al., 2021). However, we still identify problems in the estimation of decoder variance. This paper aims to obtain a better estimate of decoder variance due to suboptimal variational inference.

**Posterior collapse and KL annealing.** Posterior collapse is a prevalent issue that indicates suboptimal variational inference of VAE, where the learned encoders have a near-zero KL divergence from the prior, often due to inadequate approximation of the true posterior distribution (He et al., 2019; Lucas et al., 2019b) or, less frequently, from perfect alignment with the posterior in a degraded latent variable model Wang et al. (2021). To address this, traditional KL annealing methods reweight the loss function using a tuned hyperparameter $\beta$ (Higgins et al., 2016; Chen et al., 2018; Rezende & Viola, 2018), or a predefined monotonic or cyclical annealing schedule that involves cross-validation and intensive computation (Raiko et al., 2007; Bowman et al., 2016; Fu et al., 2019). The empirical success of these remedies can be explained by the smoothing effect of $\sigma^2$ on the loss function (Dai et al., 2021) or the information bottleneck trade-off within the KL divergence/rate term (Alemi et al., 2018; Bozkurt et al., 2021). In this paper, we provide a unified approach that integrates annealing techniques with the training dynamics of the decoder variance $\sigma^2$. Specifically, we propose an implicit $\sigma$ calibration approach that dynamically adjusts based on clear evidence of posterior collapse, leading to improved encoder estimations and reduced computational costs.

## 3 How expressive $\sigma$-CVAE can be?

A strictly positive approximation gap for all possible $\theta, \sigma$ demonstrates the inability of the CVAE generative distribution $p_{\theta,\sigma}(\boldsymbol{y}|\boldsymbol{x})$ to exactly match the data-generating distribution $p_{gt}(\boldsymbol{y}|\boldsymbol{x})$. This paper aims to understand why $\sigma$-CVAE is powerful enough to parameterize a wide range of data-generating distributions.

### 3.1 data-generating distribution and VAE generative model

We begin our analysis by defining data distributions $p_{gt}(\boldsymbol{y}|\boldsymbol{x})$ as a measurable function.

**Lemma 3.1** (**Gaussian noise outsourcing**(Agrawal & Domke, 2021)). *Suppose $\boldsymbol{x}$ and $\boldsymbol{y}$ are random vectors taking values in the standard Borel space $\mathcal{X}$ and $\mathcal{Y}$, respectively. Let $\mu_X$ denote the probability measure on $\mathcal{X}$, and let $P_{\boldsymbol{x},\boldsymbol{y}} : \mathcal{X} \times \mathcal{Y} \rightsquigarrow \mathbb{R}$ be a probability kernel of interest. Then for any $m > 1$, and a standard Gaussian random vector $\boldsymbol{z} \in \mathbb{R}^m$ with measure $\mu_Z$ that is independent of $\boldsymbol{x}$, there exists a Borel measurable function $G : \mathcal{X} \times \mathbb{R}^m \to \mathcal{Y}$ such that*

$$(\boldsymbol{x}, \boldsymbol{y}) = (\boldsymbol{x}, G(\boldsymbol{x}, \boldsymbol{z})) \quad a.s. \tag{5}$$

Lemma 3.1 states that the variability of any random variable $\boldsymbol{y}$ can be outsourced to a Gaussian random vector $\boldsymbol{z}$ that is independent of $\boldsymbol{x}$. When $\boldsymbol{x}$ is deterministic or given, such $G$ is the nested function of the quantile function of $\boldsymbol{y}$ given $\boldsymbol{x}$ and a Gaussian cumulative distribution function. Note that the dimension $m$ is not necessarily the same as the latent dimension $d$ of CVAE. Hence, for any latent $m \in \mathbb{N}^+$, there exists a Borel measurable map $\{G_m^* : \mathcal{X} \times \mathbb{R}^m \to \mathcal{Y}\}$ corresponding to the data-generating measure $\mu_{gt}$.

Based on Eq,1, we can characterize the Gaussian $\sigma$-CVAE model of $\boldsymbol{y}$ given $\boldsymbol{x}$ as a form of $G$, i.e.,

$$\boldsymbol{y} := G_{\theta,\sigma}(\boldsymbol{x}, (\boldsymbol{z}_1, \boldsymbol{z}_2)) = \mu_\theta(\boldsymbol{x}, \boldsymbol{z}_1) + \sigma \boldsymbol{z}_2, \tag{6}$$

where $\sigma > 0$. The decomposition in Eq.6 highlights the inherent restrictions of noise outsourcing in the $\sigma$-CVAE model. These restrictions include two key aspects: 1) the enforced independence between the $d$-dimensional latent variable $\boldsymbol{z}_1$ and $q$-dimensional decoder's unscaled noise $\boldsymbol{z}_2$, restricting their interactions; 2) an additive relationship between the mean and scaled variance, which limits complex interactions and constrains the flexibility of data-dependent variances.

Lemma 3.1 reformulates the approximation gap as a comparison of variable-transforming functions $G$, eliminating the need to compute log-probability and KL divergence. Such comparisons offer a new perspective on existing approximation theories. For instance, by extending and simplifying the findings of Dai & Wipf (2018) in Appendix A.2, we show that an *asymptotic* assumption on $\sigma \to 0$ eliminates the two aspects mentioned above, enabling the recovery of $G_d^*(\boldsymbol{x}, \boldsymbol{z})$ of $p_{gt}(\boldsymbol{y}|\boldsymbol{x})$ through $\mu_\theta(\boldsymbol{x}, \boldsymbol{z}_1)$ only. For another example, when the Linear VAE assumes $\mu_\theta$ as a linear combination of $\boldsymbol{x}$ and $\boldsymbol{z}$, $G_{\theta,\sigma}$ is limited to a linear function.

### 3.2 Non-asymptotic approximation of the $\sigma$-CVAE

In this section, we analyze the expressiveness of the $\sigma$-CVAE model, leveraging the above restrictions in Eq.6 in recovering the data-generating map in Eq.5. Specifically, we use a less explored yet intuitive approach known as block neural decomposition (Sobol, 2001; Märtens & Yau, 2020).

**Definition 3.2** (**Block neural decomposition** (Sobol, 2001)). Suppose that for any response dimension $q$, there exists $m > q$, such that $G_m^* : \mathcal{X} \times \mathbb{R}^m \to \mathcal{Y}$ is almost surely equal to the $f_\eta$ which can be arbitrarily complex. Then a block decomposition of $G_m^*$ on input dimension $\mathcal{X} \times \mathbb{R}^{m-q} \times \mathbb{R}^q$ is

$$\begin{aligned} f_\eta(\boldsymbol{x}, \boldsymbol{z}_1, \boldsymbol{z}_2) = &f_0^\eta + f_{\boldsymbol{x}}^\eta(\boldsymbol{x}) + f_{\boldsymbol{z}_1}^\eta(\boldsymbol{z}_1) + f_{\boldsymbol{z}_2}^\eta(\boldsymbol{z}_2) + f_{\boldsymbol{x}\boldsymbol{z}_1}^\eta(\boldsymbol{x}, \boldsymbol{z}_1) + f_{\boldsymbol{x}\boldsymbol{z}_2}^\eta(\boldsymbol{x}, \boldsymbol{z}_2) \\ &+ f_{\boldsymbol{z}_1\boldsymbol{z}_2}^\eta(\boldsymbol{z}_1, \boldsymbol{z}_2) + f_{\boldsymbol{x}\boldsymbol{z}_1\boldsymbol{z}_2}^\eta(\boldsymbol{x}, \boldsymbol{z}_1, \boldsymbol{z}_2). \end{aligned} \tag{7}$$

To be specific, this decomposition is a blockwise version of measurable functions (Sobol, 2001), making the non-parametric mixture model in Märtens & Yau (2020) a particular case. Below, we consider a unique decomposition with additional constraints on $G_m^* = f_\eta(\boldsymbol{x}, \boldsymbol{z}_1, \boldsymbol{z}_2)$.

**Assumption 3.3** (**Block ANOVA representation** (Sobol, 2001))**.** Let $\mu_x$, $\mu_{z_1}$, $\mu_{z_2}$ denote the Borel measure of $\boldsymbol{x}$, $\boldsymbol{z}_1$, $\boldsymbol{z}_2$, respectively, and let $\mu_{\mathcal{I}}$ be the measure or product measure of the index subset of $\{\boldsymbol{x}, \boldsymbol{z}_1, \boldsymbol{z}_2\}$. The block ANOVA representation assumes that the functional $f$ in Eq.7 satisfies the following integral constraints, $\int f_{\mathcal{I}}(y_{\mathcal{I}})\mu_{\mathcal{I}}(dy_i) = 0, \quad \forall i \in \mathcal{I}$, for every index subset, $\mathcal{I} \subseteq \{\boldsymbol{x}, \boldsymbol{z}_1, \boldsymbol{z}_2\}$.

So far, we have obtained a unique decomposition of the data-generating function $G_m^*$ as a sum of orthogonal functional bases, which is directly related to the oracle $\sigma$-CVAE in the form of $G_{\theta,\sigma}(X,(Z_1,Z_2))$. By establishing the equality conditions, we demonstrate below the expressiveness of Gaussian decoders, e.g., how they can have a zero approximation gap to an arbitrary data-generating map $G_m^*$, by characterizing the oracle decoders with parameters $\theta^*, \sigma^*$.

**Theorem 3.4** (**Non-asymptotic approximation of $\sigma$-CVAE**)**.** *Under Assumptions 3.3, for some $m > q$, if the block neural decomposition of the ground-truth map $G_m^*$ satisfies the following conditions almost everywhere up to zero measure of $\boldsymbol{z}_2$: 1) $f_{\boldsymbol{z}_2}(\boldsymbol{z}_2) = \sigma^* \boldsymbol{z}_2$ and 2) $f_{\boldsymbol{z}_1 \boldsymbol{z}_2}^{\eta}(\boldsymbol{z}_1, \boldsymbol{z}_2) + f_{\boldsymbol{x}\boldsymbol{z}_2}^{\eta}(\boldsymbol{x}, \boldsymbol{z}_2) + f_{\boldsymbol{x}\boldsymbol{z}_1\boldsymbol{z}_2}^{\eta}(\boldsymbol{x}, \boldsymbol{z}_1, \boldsymbol{z}_2) = C_0$ for some $\sigma^* > 0$, and constant $C_0$, then there exists a $\sigma$-CVAE model with the decoder parameterized by $\theta(\eta)$, and decoder variance scalar $\sigma$ such that*

$$G_{\theta,\sigma}(\boldsymbol{x},(\boldsymbol{z}_1,\boldsymbol{z}_2)) = G_m^*(\boldsymbol{x},\boldsymbol{z}) \quad a.s. \tag{8}$$

The proof is deferred to Appendix C. The approximation theorem 3.4 states that there exists a Gaussian decoder in $\sigma$-CVAE that well approximates any data-generating map $G_m^*$ under proper assumptions.

We avoid relying on restrictive assumptions on optimal decoder parameters $\theta^*, \sigma^*$ in Theorem 3.4, compared to existing learning theories. For example, (Dai & Wipf, 2018; Zheng et al., 2022) assumes an asymptotic condition $\sigma \to 0$ for reaching a zero approximation gap. Another line of studies (Lucas et al., 2019b; Dai et al., 2020; Sicks et al., 2021; Wang & Ziyin, 2022; Dang et al., 2023) assumes that the $G_m^*$ is highly constrained and adheres to the pPCA model, which can be parameterized by a simple linear VAE. Here, we argue that these assumptions are primarily employed to guarantee optimal variational inference. Specifically, pPCA model in linear VAEs has a closed-form conjugate normal posterior (Lucas et al., 2019b, Eq.6, page 4), and taking $\sigma \to 0$ has an asymptotic effect in reducing the inference gaps (Dai & Wipf, 2018, Eq.30, page 32). In addition, block neural decomposition of $G_m^*$ also naturally avoids the assumption that $G_m^*$ can be approximated by a finite Gaussian mixture, as used in Mattei & Frellsen (2018).

Theorem 3.4 characterizes the ideal parameters of decoders learned from optimizing ELBO, which are independent of the value of $\phi$. To this extent, it reflects and aligns with the practical limitations of $q_\phi$, such as the amortization of variational parameters and restrictions on the variational distribution family (Cremer et al., 2018). Thus, we argue that Theorem 3.4 better explains the empirical success and expressiveness of VAE models. As evident in Section 5.1, the two-moon dataset is recovered by the $\sigma$-CVAE model without $\sigma \to 0$, and it is not an instance of the pPCA model. See Appendix D for a quick comparison.

Nonetheless, Theorem 3.4 neither guarantees estimation accuracy nor offers guidance on improved training strategies for optimizing ELBO. Namely, if the inference gap varies with $\theta, \sigma$, the local optima $\hat{\theta}, \hat{\sigma}, \hat{\phi}$ shall balance the approximation gap and the inference gap, which may, however, lead to substantial deviations from the oracle parameters $\theta^*, \sigma^*$. Thus, developing practical approaches to improve the estimation of $\hat{\theta}, \hat{\sigma}$ is crucial, particularly in cases of suboptimal variational inference, including posterior collapse.

## 4 What if variational inference is not optimal?

The rest of this work addresses the challenges in optimizing the ELBO objective posed by suboptimal variational inference. In particular, we provide a fine-grained analysis of the duality between the dynamics of $\sigma$ and $\phi$, uncovering how they may contribute to the identified issues. This dual relationship is the primary motivation for a unified annealing technique, which eliminates poor estimates $\hat{\theta}, \hat{\sigma}$ to enhance the CVAE's generative performance in general.

### 4.1 Biased estimation: Suboptimal inference induced saddle points

Traditional wisdom in likelihood-based models has been optimizing the loss by updating $\sigma$ jointly with $\{\theta, \phi\}$ (Dai et al., 2021; Dai & Wipf, 2018) or by updating $\sigma$ iteratively (Rybkin et al., 2021) to avoid unbounded likelihood or numeric instability at *small* $\sigma$, where the error of $\theta$ is magnified, and the gradient of $\sigma$ or $\mu$ explodes (Mattei & Frellsen, 2018; Takahashi et al., 2018).

In this section, we analyze the training dynamics of $\sigma$, showing how they could lead to a significant source of numerical instability (saddle points) in optimizing the ELBO in a different phenomenon.

Recalling Fisher's identity, the gradient of $L(\theta, \phi, \sigma)$ with respect to $\sigma$ is a biased *approximation* of the true gradient of the negative log-likelihood function with respect to $\sigma$, where $q_\phi$ controls the bias.

$$\nabla_\sigma L(\theta, \phi, \sigma) = \mathbb{E}_D \mathbb{E}_{q_\phi}[-\nabla_\sigma \log p_{\theta,\sigma}(\boldsymbol{y}|\boldsymbol{x}, \boldsymbol{z})] \neq \mathbb{E}_D \mathbb{E}_{p_{\theta,\sigma}(\boldsymbol{z}|\boldsymbol{y}, \boldsymbol{x})}[\nabla_\sigma \log p_{\theta,\sigma}(\boldsymbol{y}|\boldsymbol{x}, \boldsymbol{z})]; \tag{9}$$

Similarly, the estimate $\sigma_t$ given $\{\theta_t, \phi_t\}$ in the two-step iterative method in Rybkin et al. (2021), obtained by maximizing ELBO analytically, is

$$\sigma_t^2 = \arg\min_\sigma L(\theta_t, \phi_t, \sigma) = \mathbb{E}_D \mathbb{E}_{q_{\phi_t}} \left[(\boldsymbol{y} - \mu_{\theta_t}(\boldsymbol{x}, \boldsymbol{z}))^2\right] \neq \mathbb{E}_D \mathbb{E}_{p_{\theta_t, \sigma_{t-1}}(\boldsymbol{z}|\boldsymbol{y}, \boldsymbol{x})} \left[(\boldsymbol{y} - \mu_{\theta_t}(\boldsymbol{x}, \boldsymbol{z}))^2\right]. \tag{10}$$

Here, we highlight that the suboptimality of $q_\phi$ is driven more by the effect of amortized inference than by the constraints of the distribution family. Furthermore, $q_\phi$ more easily recovers the posterior mean than approximates the posterior variance (Cremer et al., 2018). Thus, $q_\phi$ is less peaked and less concentrated than the posterior, and the estimated $\sigma$ is often larger than its actual value (Lin et al., 2019). In Appendix E, we also prove a toy example of inflated estimates of $\sigma$ via Fisher's identity.

As shown in Fig.1 of Lucas et al. (2019b), we argue that an inflated $\sigma$ may lead to increased saddle points in the loss landscape, contrary to the ideal scenario where optimal variational inference is achievable, as seen in the asymptotic analysis (Dai & Wipf, 2018; Zheng et al., 2022) or in a linear VAE setting (Lucas et al., 2019b; Dai et al., 2020; Wang & Ziyin, 2022). Such saddle points increase the sensitivity of the optimization process to the initialization of $\sigma$, as shown in Lucas et al. (2019a). Thus, additional efforts are needed for numerical stability and accurate estimation of $\sigma$, especially if the decoder covariance matrix is flexible.

### 4.2 Posterior collapse: a compelling evidence for suboptimal encoders

In this section, we demonstrate the smoothing effect of $\sigma$ in preventing certain $q_\phi$, highlighting the opposite direction of influence. We argue that actions should also be taken if there is clear and compelling evidence of a poor encoder $q_\phi$. Specifically, we consider a well-known example of the evidence, termed *posterior collapse* (Lucas et al., 2019a; He et al., 2019; Razavi et al., 2019) or *KL vanishing* (Bowman et al., 2016; Fu et al., 2019). It refers to the problem that the approximate posterior $q_\phi(\boldsymbol{z}|\boldsymbol{y}, \boldsymbol{x})$ collapses to the standard Gaussian $N(0, I_d)$, resulting in vanishing KL divergence in the training loss function.

$$\mathbb{E}_D \mathrm{KL}[q_\phi(\boldsymbol{z}|\boldsymbol{y}, \boldsymbol{x})||p(\boldsymbol{z})]] \approx 0. \tag{11}$$

The encoder $q_\phi$ approximating the true posterior will hardly become the prior. This is because under Bayes' rule, $p_{\theta,\sigma}(\boldsymbol{z}|\boldsymbol{y}, \boldsymbol{x}) \propto p(\boldsymbol{z})p_{\theta,\sigma}(\boldsymbol{y}|\boldsymbol{x}, \boldsymbol{z})$ and $p_{\theta,\sigma}(\boldsymbol{y}|\boldsymbol{x}, \boldsymbol{z})$ will not be independent of $\boldsymbol{z}$. In our setting, the decoder's mean $\mu_\theta(\boldsymbol{x}, \boldsymbol{z})$ will not be independent of $\boldsymbol{z}$. If the VAE generative model ignores a latent variable, it collapses to nonlinear regression with a mean network $\mu_\theta(\boldsymbol{x})$, significantly reducing the model's expressive power and the complexity of the noise structure. As a result, posterior collapse often coincides with poor ELBO in complex datasets due to a lack of fitness.

Traditionally, the standard solution has been KL annealing (Raiko et al. (2007); Bowman et al. (2016); Fu et al. (2019)) leveraging $\beta$-VAE. We refer to it as explicit $\beta$-annealing. The idea is that the loss landscape w.r.t. $\theta, \phi$ is highly non-convex when $\sigma$ is fixed. Thus, tuning the hyperparameter $\beta$, which reweights the KL divergence term in the loss function, results in a relatively large KL divergence in Eq.11 and a significantly lower training ELBO loss. For example, the schedule of $\beta$ in Fu et al. (2019) can be

$$\beta_t = 2r/M * 1_{\{0 \leq r < M/2\}}(r) + 1 * 1_{\{M/2 \leq r < M\}}(r), \tag{12}$$

---

**Algorithm 1**: general $\sigma$ calibration for $\sigma$-CVAE

---

**Input:** data $\{\boldsymbol{x}_i, \boldsymbol{y}_i\}_{i=1}^N$, deterministic neural networks $f_{\mu_y}, f_{\mu_z}, f_{S_y}$ with initialized parameter $\theta_0, \phi_0$, Initialization $\sigma_0$, inner number of iteration $K$ of updating $\theta, \phi$, Calibration tolerance $C$

**Output:** $\theta, \phi, \sigma$

**while** is training **do**

  **for** $i = 1$ **to** $K$ **do**

    Read batch $\{\boldsymbol{x}_j, \boldsymbol{y}_j\}_{j=1}^B$ from data

    Sample $\boldsymbol{z}_j \leftarrow q_{\phi_t}(\boldsymbol{z}|\boldsymbol{y}_j, \boldsymbol{x}_j)$ for $j = 1, ..., B$

    Compute batch loss $L(\theta_t, \phi_t, \sigma_t)$

    Compute gradient $\nabla_\theta L; \nabla_\phi L$

    $\theta_t \leftarrow \theta_t - \alpha \nabla_\theta L$

    $\phi_t \leftarrow \phi_t - \alpha \nabla_\phi L$

  **end for**

  Sample $\boldsymbol{z}_j \leftarrow q_{\phi_t}(\boldsymbol{z}|\boldsymbol{y}_j, \boldsymbol{x}_j)$ for $j = 1, ..., N$;

  Update $\sigma_t^2 \leftarrow \frac{1}{N} \sum_{i=1}^N \|\boldsymbol{y}_i - \mu_{\theta_t}(\boldsymbol{x}_i, \boldsymbol{z}_i)\|^2 / q$

  **if** Convergence **then**

    **if** $\sum \text{KL}[q_\phi(\boldsymbol{z}|\boldsymbol{y}_i, \boldsymbol{x}_i)||p(\boldsymbol{z})] < C$ **then**

      Calibrate $\sigma_t \leftarrow \sigma_t^{cal}$ as Eq.15

    **end if**

  **else**

    Break

  **end if**

**end while**

---

where $r = \text{mod}(t, M)$ is the remainder of time $t$ divided by the length of the annealing cycle $M$. In other words, $\beta_t$ is periodically annealed from 0 to 1 during the first half of a $M$-iteration cycle and remains at 1 throughout the rest of the cycle.

In this paper, we highlight the mathematical equivalence between $\sigma$ and $\beta$ in Eq.4. In probabilistic modeling terms, $\beta$-annealing can be viewed as a sequence of *pre-defined* and *fixed* calibration of scaled variance $\sigma$. Taking Eq.12 as an example, the loss-equivalent decoder variance $\sigma_t$ is annealed in the same $r$, i.e.,

$$\sigma_t^2 = M/4r * 1_{\{0 \leq r < M/2\}}(r) + 1/2 * 1_{\{M/2 \leq r < M\}}(r). \tag{13}$$

Perhaps surprisingly, the equivalence reveals the subtlety of the hyperparameter $\beta$. From a statistical perspective, when the decoder is taken from a location-scale distribution family, scaling the KL divergence (KLD) in ELBO is nothing but scaling its unit diagonal variance.

Thus, several aspects of $\beta$-annealing are ad hoc. First, the assumption of a fixed $\sigma$ limits the expressive power of the marginal distribution $p_{\theta,\sigma}(\boldsymbol{y}|\boldsymbol{x})$, causing a potentially larger approximation gap. Secondly, the training dynamics of $\sigma$ can hinder the annealing effect in shaping the loss landscape due to such equivalence (See Appendix G.1 for a detailed discussion). In addition, the $\beta$-scaled objective lacks statistical interpretability if $\beta \neq 1$ since it can no longer be seen as an approximate log-likelihood. Lastly, the insight of the annealing schedule is unclear in principle, let alone the tuning of its hyperparameter, e.g., how to choose the period $M$ for a better model. From an optimization theory perspective, preventing posterior collapse can be easily achieved without introducing $\beta$ because annealing $\beta$ is equivalent to $\sigma$ in changing the loss landscape of $\theta, \sigma$. This statement remains valid even if $\sigma$ is neither learned nor optimal. However, treating $\sigma$ as a model parameter is essential for achieving improved probabilistic modeling and capturing uncertainty more effectively.

To conclude, we have demonstrated the whole picture of the duality of $\sigma$ and $\phi$. Suboptimal $q_\phi$ leads to a biased and inflated estimation of $\sigma$, causing additional saddle points in optimizing the ELBO. On the other hand, the smoothing effect of $\sigma$ alleviates poor $q_\phi$, thereby mitigating posterior collapse.

### 4.3 $\sigma$-calibration: an implicit KL annealing with a learned $\sigma$

In this paper, we propose Calibrated Robust $\sigma$-CVAE, a simple framework incorporating calibrations of $\sigma$ using the block coordinate gradient descent method (Rybkin et al., 2021). We consider the optimization problem:

$$\min_{\theta,\sigma,\phi} \mathbb{E}_D[-\text{ELBO}(\theta, \sigma, \phi)] \quad \text{subject to: } \mathbb{E}_D[\text{KL}[q_\phi(z|y, x)||p(z)] \geq C. \tag{14}$$

Calibrated Robust $\sigma$-CVAE extends Rybkin et al. (2021) with two key enhancements and operates through the following three steps in each iteration. A general pseudo-code can be found in Algorithm 1.

i. **Optimize** $\theta, \phi$: Perform $K$ stochastic gradient steps for $\theta, \phi$ with fixed $\sigma$, where $K$ is the *inner steps* size.

ii. **Estimate** $\sigma$: Compute the best estimate of $\sigma$ analytically based on current $\theta, \phi$ (Rybkin et al., 2021).

iii. **Calibrate $\sigma$ at convergence**: At the convergence of ELBO, calibrate $\sigma$ if $\mathbb{E}_D[\text{KL}[q_\phi(z|y,x)||p(z)] < C$.

Specifically, if the KL divergence term in the loss (KLD, or equivalently Rate (Alemi et al., 2018)) is less than the hyperparameter *calibration tolerance* $C$, the calibrated $\sigma^{\text{cal}}$ is defined as:

$$\sigma_t^{\text{cal}} := |1 + \varphi| \times \sigma_t, \tag{15}$$

where $\varphi \sim N(0, S)$ is sampled from a normal distribution with variance $S$. The variance $S$ starts at 1 and increases by 1 after each calibration. An example of the calibration process of $\sigma$ is shown in Figure 2(a). The constraint that KLD $\geq C$ defines the feasible region of $\phi$ in the constrained optimization of ELBO. It excludes any local optima $\theta, \sigma, \phi$ in ELBO such that this constraint is not satisfied.

In recognition of the aforementioned issues, the merit of calibration is supported by the following rationales:

1) In the context of annealing, calibration is an annealing that is adaptive to the performance of $\sigma$-CVAE, where the rate term is a representative performance metric that is directly reflected in the optimization problem as a hard constraint (See the discussion below). Our calibration is similar to the sudden change of $\beta$ at the end of a cycle (Fu et al., 2019). The annealing effect in our algorithm is concentrated, and the stage of slow annealing in $\beta$-annealing is replaced by the learning dynamics of minimizing $\sigma$ as a learned parameter. $C$ is the minimal desired KLD after annealing. Note that the relationship between $C$ and hyperparameters in $\beta$-annealing is not bijective (See Fig.12 in Fu et al. (2019)).

2) To mitigate the numerical instability described in Section 4.1, our proposed calibration follows the perturbation and random restarting strategies of $\sigma$ to navigate saddle points. Specifically, $\varphi$ governs the calibration as scaling adjustments. The adjustment strength (annealing temperature) is controlled by the variance $S$ of $\varphi$. Due to the above concentrated annealing effect, we may require larger calibrations. Its escalating feature of variance $S$ is beneficial when larger calibration is necessary to escape the flat regions of the loss landscape.

3) Calibration is simple and straightforward—no calibration on $\sigma$ results from the local optima of ELBO, and a predefined annealing schedule is no longer needed. It has minimal impact on model interpretability since our primary objective is the original ELBO loss and avoids the explicit usage of $\beta$. Due to the nature of the constraint and non-convexity, the learned $\theta, \sigma, \phi$ is highly likely to be a non-boundary solution.

Finally, we demonstrate the flexibility of our method and a better understanding of the interpretation of KLD and the optimum values of the hyperparameter $C$. 1) To prevent posterior collapse, KLD is the direct metric, and the optimal $C$ is any value between 0 and the lowest KLD of $q_\phi$ that is not posteriorly collapsed. In practice, we define posterior collapse first by considering the permissible deviation of a zero KLD. Such permissible deviation, which becomes the optimal $C$, should be slightly more than negligible and may be subjective. 2) To fine-tune the model for various tasks, the metric of KLD/Rate is the preference and/or strategy for selecting from nearly the best local optima $\phi$, particularly when multiple distinct local optima of the ELBO yield approximately the same minimal training loss. For example, in the rate-distortion trade-off (Alemi et al., 2018), $C$ corresponds to the minimum value of the upper bound on mutual information, which can be tuned to improve latent disentanglement. For another example, in the approximation-inference trade-off in Eq.3, $C$ is chosen to prioritize one of the gaps for better generative or inference performance. Here, the optimal $C$ is objectively unknown and must be carefully tuned for task-specific goals.

## 5 Experiments

In this section, we provide evidence and more practical insights into the calibration of $\sigma$-CVAE through extensive numerical experiments given a finite training sample. We use the Adam (Kingma, 2014) stochastic gradient descent algorithm for training neural networks. The general learning rate is 0.005, and the convergence threshold is 0.001 in the average loss change. Other important details of each of the following experiments are attached in the Appendix F. In addition, we also validate our method on the MNIST (Deng, 2012) and CelebA (Liu et al., 2018) datasets for conditional image generation and reconstruction in Appendix I & J.

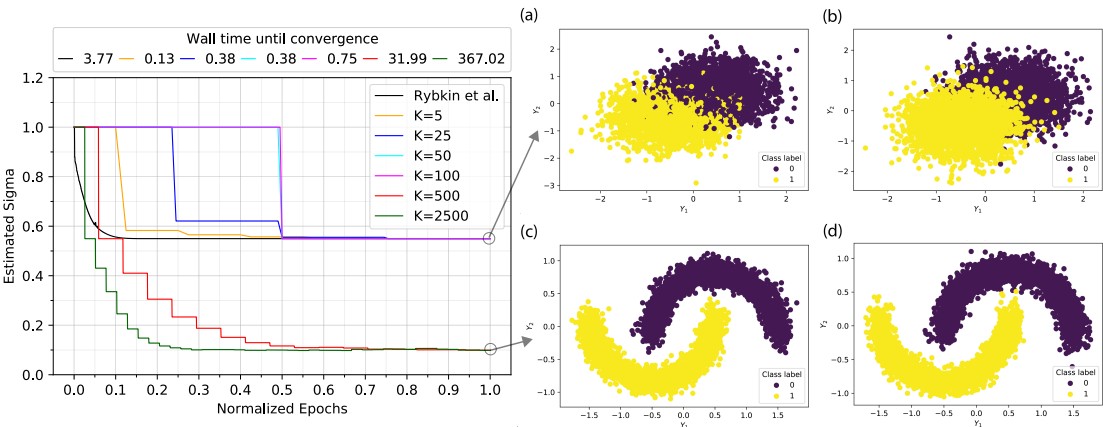

Figure 1: Trajectories of $\sigma$ estimated under different values of $K$. Dynamic scaling on total epochs is applied. (a),(c) reconstructed training Two-moon dataset; (b),(d) generated samples. Vanilla $\sigma$-CVAE may not recover the true $\sigma$, reconstruct training data, and learn the ground-truth distribution. The true $\sigma$ is 0.1.

## 5.1 Two-moon dataset

Let $\boldsymbol{x} \in \{-1, 1\}$ be the binary input and let $\boldsymbol{y} \in \mathbb{R}^2$ be generated as follows,

$$\boldsymbol{y} = \begin{cases} \left(\cos(\alpha) + \frac{1}{2} + \epsilon_1, \sin(\alpha) - \frac{1}{6} + \epsilon_2\right), & \text{if } \boldsymbol{x} = -1, \\ \left(\cos(\alpha) - \frac{1}{2} + \epsilon_3, -\sin(\alpha) + \frac{1}{6} + \epsilon_4\right), & \text{if } \boldsymbol{x} = 1, \end{cases}$$

where latent $\alpha \sim \text{Uniform}[0, \pi]$ and $\epsilon_i \sim N\left(0, \tau^2\right), \forall i = 1, 2, 3, 4$ are mutually independent. We simulated three datasets of size $n = 5,000$ with $2,500$ for each class using $\tau = 0.1$, which is referred to as **the true** $\sigma$. Note that this synthetic dataset does not follow the pPCA model described in Lucas et al. (2019a) and cannot be recovered by a linear CVAE. A well-behaved encoder, in this case, is expected to yield a non-zero KL divergence to the prior. We have the following observations.

**No calibration: a properly learned decoder variance is crucial for probabilistic modeling.** In Figure 1, we first explore the vanilla method that does not include the third step, including $K = 1$ as suggested in Rybkin et al. (2021). We see that different inner steps $K$ result in different generative models at convergence, given the same training dataset. The contradictory result is well predicted by Theorem 3.4— although $\sigma$-CVAE is not guaranteed to achieve optimal variance inference. $p_{\theta,\sigma}$ in Eq.6 may well approximate the data-generating distribution in Eq.5. However, we also see that the vanilla $\sigma$-CVAE under a small value of $K$, including $K = 1$, fails to recover the truth. This observation agrees with the claims that learning $\sigma$ can have unintended consequences (Lucas et al., 2019b; Dai et al., 2021), highlighting practical difficulties in parameter estimation. The observation that increasing the inner step size $K$ improves the performance of the generative model aligns with our discussion in Section 4.1, where the bias of $\sigma$ can be reduced by obtaining a more informative $\phi$. While a smaller $K$ is generally preferred for computational efficiency, the appropriate $K$ is generally unknown and differs from case to case. Finding the right balance between parameter estimation and computational efficiency remains a non-trivial challenge.

**Evaluating the effect of calibration as a third step.** In Figure 2, we evaluate the effect of calibration as the third step, compared to the vanilla method. Calibrated robust $\sigma$-CVAE accurately estimates the decoder variance in Figure 2(a). In Figure 2(b), it prevents posterior collapse and obtains a better generative model with a low negative ELBO. Compared with the vanilla CVAE that recovers the true $\sigma$ using a large $K = 500, 2500$, our algorithm using $K = 25$ requires a significantly smaller wall time to converge. Table 6 in Appendix H compares the wall time until convergence for different setups of $K$.

**Compare with KL $\beta$-annealing.** In Table 1, we compare Monotonic (Bowman et al., 2016) and Cyclical annealing (Fu et al., 2019) with our method. We report the MSE of the learned $\sigma$ and KLD at convergence, under the same setting as Figure 3 averaging over 20 repetitions. Training loss is not reported due to the inconsistency of the loss reweighted by $\beta$. We found that by using a default small $C = 0.05$, our algorithm

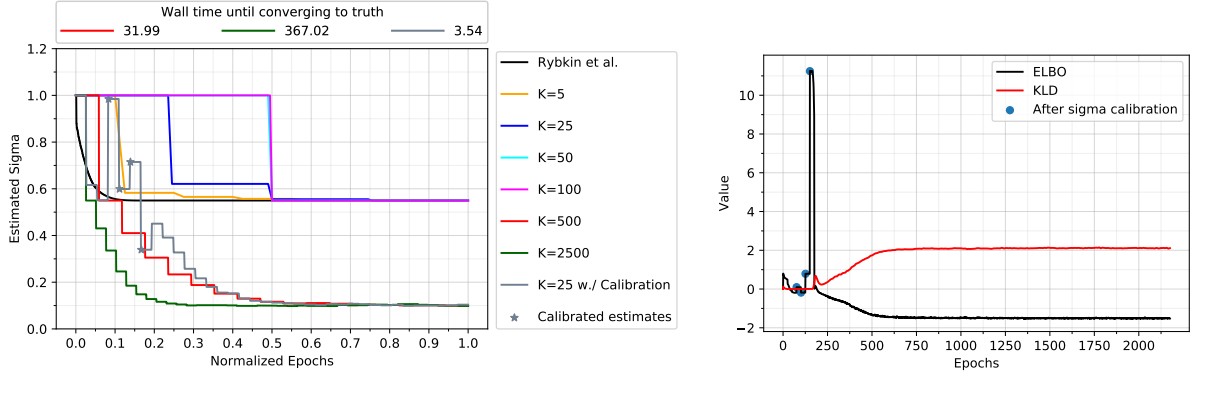

(a) $\sigma$-calibration in a small $K$ setting

(b) $\sigma$-calibration is equivalent to KL annealing

Figure 2: (a) The trajectory of the $\sigma$ estimated by our proposed calibration technique. Accurate estimate of $\sigma$ is obtained by calibration with a small $K$ (K=25 w./ Calibration). The calibrated $\sigma$ are marked as $*$; (b) The KL divergence in ELBO remains positive after several calibrations of $\sigma$. The calibration tolerance $C = 0.05$. The true $\sigma$ is 0.1.

Table 1: Comparison of KLD and MSE of the learned $\sigma$ across different annealing techniques

| $\sigma$ Initialized | 0.2 | | 0.4 | | 0.6 | | 0.8 | | 1 | |
|---|---|---|---|---|---|---|---|---|---|---|
| | MSE | KLD | MSE | KLD | MSE | KLD | MSE | KLD | MSE | KLD |
| Monotonic | 1.62e-02 | 1.81 | 6.09e-03 | 2.10 | 5.65e-03 | 2.19 | 3.41e-04 | 2.12 | 6.06e-04 | 2.17 |
| Cyclical | 1.76e-02 | 1.77 | 3.24e-02 | 1.71 | 1.92e-02 | 1.84 | 2.17e-02 | 2.01 | 4.58e-02 | 1.45 |
| Ours C=0.05 | 2.41e-02 | 1.79 | 2.71e-02 | 1.64 | 6.01e-03 | 1.92 | 5.38e-02 | 1.39 | 2.69e-02 | 1.70 |
| Ours C=1.96 | **1.26e-05** | 2.09 | **5.77e-05** | 2.09 | **1.22e-05** | 2.08 | **1.73e-05** | 2.10 | **1.76e-05** | 2.09 |
| Vanilla | 0.16 | 4.17e-01 | 0.20 | 1.40e-03 | 0.17 | 3.16e-01 | 0.20 | 2.20e-04 | 0.20 | 5.39e-04 |

Table 2: Comparison of wall time to convergence across different annealing techniques.

| Inner step size K | 25 | 50 | 100 |
|---|---|---|---|
| Monotonic | 24.85 | 36.26 | 58.38 |
| Cyclical | 10.76 | 23.01 | 45.59 |
| Ours C=0.05 | **10.55** | **12.69** | **25.00** |
| Vanilla | — | — | 62.15 |

results in a similar KLD and similar MSE; however, our algorithm can be $50 - 70\%$ faster to converge in Table 2, defined as the maximum wall time until convergence. In addition, we fine-tune our algorithm by adjusting $C = 1.96$, leading to a significantly less biased estimation of the decoder. A detailed comparison of wall time savings and other experimental details is in the Appendix G.

**Small initialization strategy and ablation study.** In Figure 3, we report the results of Calibrated Robust $\sigma$-CVAE with various *initializations* of $\sigma$ ranging from 0.05 to 1. In Figure 3(b), Calibrated Robust $\sigma$-CVAE demonstrates consistently superior performance in various metrics. In Figure 3(a), the vanilla CVAE ends up with a very high training loss (negative ELBO), a large mean squared error (MSE) of the learned $\sigma$, and a vanishing KLD. Similar to Dai & Wipf (2018); Dang et al. (2023), we find that a smaller initialization of $\sigma$ has a higher chance of avoiding posterior collapse with a non-zero KLD. However, the strategy of smaller initializations is limited, and it may still fail to recover the ground-truth distribution. In Figure 3(b), Calibrated Robust $\sigma$-CVAE demonstrates consistently superior performance in various metrics. In the ablation study in Appendix H, we consider various setups of $C$ from 0 to 0.5, where $C = 0.05$ is effective enough for preventing posterior collapse and providing an accurate estimate of $\sigma$. Results of a *fixed-$\sigma$* CVAE under different initializations can also be found in Appendix H. If the fixed $\sigma$ happens to recover the true $\sigma = 0.1$, it yields the best result.

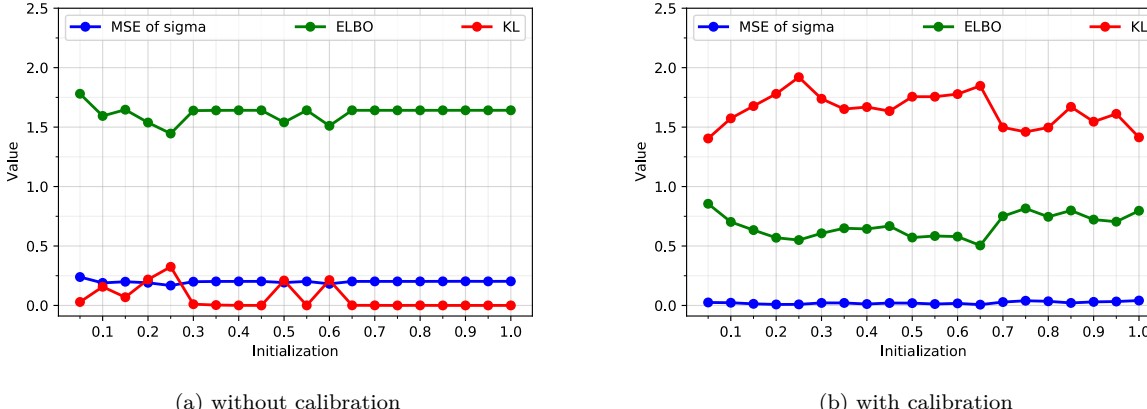

(a) without calibration                                     (b) with calibration

Figure 3: CVAE model using a learned $\sigma$ (a) without calibration and (b) without our proposed calibration on twomoon training datasets. It prevents a vanishing KL divergence due to posterior collapse and significantly improves the decoder variance estimates and the ELBO at convergence. The true $\sigma = 0.1$. The calibration tolerance $C = 0.05$. Inner step $K = 25$. Each point is averaged over 20 repetition experiments.

## 5.2 Comparison with nonparametric conditional density estimator

Next, we compare the Wasserstein generative conditional sampling method (WGCS) (Liu et al., 2021), along with three conventional nonparametric conditional density estimators, including the nearest-neighbor kernel conditional density estimator (NNKCDE) (Dalmasso et al., 2020), the conditional kernel density estimator (CKDE) (Hall et al., 2004), and a basis expansion method (FlexCode) (Izbicki & Lee, 2017) in the tasks of estimating summary statistics including the conditional mean and conditional standard deviation. The metric for the generative performance of $\theta, \sigma$ is the mean squared error (MSE) of the estimated mean (MEAN) and that of the estimated standard deviation (SD). A smaller MSE of the estimated SD means a more accurate estimate of variance. Three different simulated datasets are used, where $M_1$ is a nonlinear model with additive noise; $M_2$ is a nonlinear model with multiplicative noise, and $M_3$ is a two-component Gaussian Mixture Model. The dimensions of label $M_1$ and $M_2$ are 5. The dimension of label $M_3$ is 2. The dimension of data $Y$ is 1. Details can be found in Appendix F. Our method produces consistent results and outperforms the other models, as shown in Table 3. The numbers in bold indicate the best results in the same row. The evaluation is repeated 10 times.

Table 3: Comparison of summary statistics between non-parametric conditional density estimators

|       |      | CVAE (OURS)        | WGCS        | NNKCDE      | CKDE         | FLEXCODE     |
|-------|------|--------------------|-------------|-------------|--------------|--------------|
| $M_1$ | MEAN | **0.174** $(0.004)$ | $1.10(0.05)$ | $2.49(0.01)$ | $3.30(0.02)$  | $2.30(0.01)$  |
|       | SD   | **0.185**$(0.005)$  | $0.24(0.04)$ | $0.43(0.01)$ | $0.59(0.01)$  | $1.06(0.08)$  |
| $M_2$ | MEAN | **0.421**$(0.005)$  | $3.71(0.23)$ | $6.09(0.07)$ | $66.76(2.06)$ | $10.20(0.33)$ |
|       | SD   | **2.071**$(0.019)$  | $3.52(0.17)$ | $9.33(0.23)$ | $18.87(0.59)$ | $11.08(0.34)$ |
| $M_3$ | MEAN | **0.106**$(0.001)$  | $0.32(0.03)$ | $0.11(0.01)$ | $1.55(0.03)$  | $0.12(0.04)$  |
|       | SD   | $0.421(0.001)$      | **0.10**$(0.01)$ | $0.36(0.01)$ | $0.51(0.01)$  | $0.33(0.01)$  |

## 5.3 Comparison with Conditional Flow Based model in benchmark datasets

Finally, we compare the performance of calibrated $\sigma$-CVAE with Bayesian conditional normalizing flows (Trippe & Turner, 2018) on 6 UCI datasets, as listed in Table 4. The test mean log-likelihood in the natural unit of information (nats) is evaluated with the associated variance reported. The models include Bayesian normalizing flows (NF), mixture density networks (MNM), and neural networks with latent inputs mean field Gaussian (MF), sampled from Hamiltonian Monte Carlo (HMC). A higher logarithmic likelihood

Table 4: Comparison of mean log-likelihood of test data in nats between conditional models.

| Dataset | boston | concrete | energy | power | wine-red | yacht |
|---|---|---|---|---|---|---|
| N | 506 | 1030 | 768 | 9568 | 1599 | 308 |
| $p/q$ | 13/1 | 8/1 | 8/2 | 4/1 | 11/1 | 6/1 |
| MDN-2 | $-2.65 \pm 0.03$ | $-3.23 \pm 0.03$ | $-1.60 \pm 0.04$ | $-2.73 \pm 0.01$ | $-0.91 \pm 0.04$ | $-2.70 \pm 0.05$ |
| MDN-5 | $-2.73 \pm 0.04$ | $-3.28 \pm 0.03$ | $-1.63 \pm 0.06$ | $-2.70 \pm 0.01$ | $\mathbf{+1.43 \pm 0.07}$ | $-2.54 \pm 0.10$ |
| MDN-20 | $-2.74 \pm 0.03$ | $-3.27 \pm 0.02$ | $-1.48 \pm 0.04$ | $-2.68 \pm 0.01$ | $+1.21 \pm 0.06$ | $-2.76 \pm 0.07$ |
| LV-15 | $-2.64 \pm 0.05$ | $-3.06 \pm 0.03$ | $-0.74 \pm 0.03$ | $-2.81 \pm 0.01$ | $-0.98 \pm 0.02$ | $-1.01 \pm 0.04$ |
| LV-5 | $-2.56 \pm 0.05$ | $-3.08 \pm 0.02$ | $-0.79 \pm 0.02$ | $-2.82 \pm 0.01$ | $-0.96 \pm 0.01$ | $-1.15 \pm 0.05$ |
| NF-2 | $-2.40 \pm 0.06$ | $-3.03 \pm 0.05$ | $-0.44 \pm 0.04$ | $-2.73 \pm 0.01$ | $-0.87 \pm 0.02$ | $-0.30 \pm 0.04$ |
| NF-5 | $-2.37 \pm 0.04$ | $-2.97 \pm 0.03$ | $-0.67 \pm 0.15$ | $\mathbf{-2.68 \pm 0.01}$ | $-0.76 \pm 0.10$ | $\mathbf{-0.21 \pm 0.09}$ |
| HMC | $\mathbf{-2.27 \pm 0.03}$ | $\mathbf{-2.72 \pm 0.02}$ | $-0.93 \pm 0.01$ | $-2.70 \pm 0.00$ | $-0.91 \pm 0.02$ | $-1.62 \pm 0.02$ |
| Dropout | $-2.46 \pm 0.25$ | $-3.04 \pm 0.09$ | $-1.99 \pm 0.09$ | $-2.89 \pm 0.01$ | $-0.93 \pm 0.06$ | $-1.55 \pm 0.12$ |
| MF | $-2.62 \pm 0.06$ | $-3.00 \pm 0.03$ | $-0.57 \pm 0.04$ | $-2.79 \pm 0.01$ | $-0.97 \pm 0.01$ | $-1.00 \pm 0.10$ |
| CVAE(ours) | $-3.17 \pm 0.07$ | $-3.48 \pm 0.04$ | $\mathbf{+0.22 \pm 0.10}$ | $-3.29 \pm 0.03$ | $-1.48 \pm 0.06$ | $-0.34 \pm 0.02$ |

implies a better generalization of the test set. The random train-test split is 75% to 25%. Each experiment is repeated 5 times. Although $\sigma$-CVAE optimizes the ELBO instead of the exact log-likelihood and does not have an optimal VI, our results are competitive. It shows superior performance on the Energy dataset, and near-optimal performance on the Energy dataset, and near optimal performance on the Yacht dataset.

# 6 Discussion

**Limitation on approximation gap:** Our analysis and algorithm setup are subject to several conventional assumptions. These assumptions can be relaxed in several aspects, including generalizing our analysis to accommodate a heterogeneous, diagonal, or even full covariance matrix $\Sigma$, which offers better approximation properties for more complex datasets.

**Other calibration metrics:** Although the KLD or Rate in Eq.11 is a direct measurement, the criteria for posterior collapse remain flexible. For example, Lucas et al. (2019b) define the dimension-wise posterior collapse by $\mathrm{P}(\mathrm{KL}[q_\phi(\boldsymbol{z}_i|\boldsymbol{y},\boldsymbol{x})||p(\boldsymbol{z}_i)] \leq \epsilon) > 1 - \delta$ for each latent dimension index $i$ in a probably approximately correct manner. For another example, Dai et al. (2021) define implicit posterior collapse from observing a large maximum mean discrepancy (Makhzani et al., 2016) between the aggregated posterior $\frac{1}{N}\sum_{i=1}^{N} q_\phi(\boldsymbol{z}|\boldsymbol{y}_i,\boldsymbol{x}_i)$ and $p(\boldsymbol{z})$. This is because marginalizing a well-behaved encoder would be similar to the prior: $\int q_\phi(\boldsymbol{z}|\boldsymbol{y},\boldsymbol{x})\mu_{gt}(d\boldsymbol{y}d\boldsymbol{x}) \approx \int p_{\theta,\sigma}(\boldsymbol{z}|\boldsymbol{y},\boldsymbol{x})\mu_{gt}(d\boldsymbol{y}d\boldsymbol{x}) \approx p(\boldsymbol{z})$.

**Hyperparameter selection:** Calibrated Robust $\sigma$-CVAE depends on two hyperparameters, namely $C, K$. While it eliminates the need for a predefined annealing schedule (Raiko et al., 2007; Bowman et al., 2016; Fu et al., 2019), these hyperparameters are crucial and must be carefully tuned. We should generally use a small $K$ and set $0 < C < H(Y|X)$. We highlight that a higher KLD does not necessarily mean a better $q_\phi$ or a lower inference gap, and the trade-off for the near-optimal solutions is task-specific and date-dependent. As a general rule of thumb, we recommend using a small tolerance $C$ and moderate $K$ inner steps as a starting value for computational efficiency. Choosing an enormous tolerance $C$ will not only trigger more calibrations of $\sigma$ but also lead to potential non-convergence of the algorithm.

**Image generation tasks:** If probabilistic modeling and parameter estimation are not the top priorities of the task, it is expected to use a fixed $\sigma$ or incorporate both $\beta \neq 1$ and a learned $\sigma$, especially in image generation tasks, including crisp face generation (Vahdat & Kautz, 2020) and abnormality detection (Loizillon et al., 2024). Based on results from Appendix I & J, $\sigma$-calibration itself does not significantly improve the image generation results. The image generation process often deviates from the complete generative graph, as the decoder's mean is used directly as the generated sample. It neither aligns with the assumptions of a Gaussian decoder nor accurately represents the underlying generative process.

## 7    Conclusion

To the best of our knowledge, this work represents the first comprehensive investigation of the variance parameter $\sigma$ in VAE models. In this work, our methodology provides deeper insight into the approximation power of VAE models and the duality between $\sigma$ and $\phi$, which enables a novel pathway for $\sigma$-annealing. We empirically demonstrate the effectiveness of our proposed Calibrated Robust $\sigma$-CVAE in addressing the challenges of accurate decoder variance estimation, numerical instabilities due to suboptimal variational inference, and posterior collapse. Our future work will focus on generalizing our approach to encompass broader and more complex issues in conditional learning, particularly those involving unbalanced sampling and missing data.

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

# A Discussion on Theorem 3.4

The approximation theories of this work are also motivated by explaining the expressiveness of the parameterized density $p_{\theta,\sigma}(y|x)$ when $\sigma$ is not infinitesimal. Here, we show a simple extended result from Dai & Wipf (2018).

**Assumption A.1** (**Response continuity**). Conditional on any $\boldsymbol{x} \in \mathcal{X}$, there exists a ground truth probability measure of $\mu_{gt}$ such that its Radon-Nikodym derivative with respect to the standard Lebesgue measure is nonzero almost everywhere in $\mathcal{Y}$.

Simply put, $p_{gt}(\boldsymbol{y}|\boldsymbol{x})$ exists uniquely and is non-zero almost everywhere up to a null set.

**Proposition A.2** (**Global optimal $\sigma$-CVAE**). *Under Assumption A.1, if the latent dimension $d$ is larger than or equal to the dimension of the response $q$, then for any $\sigma > 0$, there exists a $\sigma$-CVAE with encoder network parameterized by $\theta_\sigma$, and decoder network parameterized by $\phi_\sigma$ such that,*

$$\lim_{\sigma \to 0} \mathbf{KL}[p_{\theta_\sigma}(y|x)\|p_{gt}(y|x)] = 0, \lim_{\sigma \to 0} \mathbf{KL}[q_{\phi_\sigma}(z|y,x)\|p_{\theta_\sigma}(z|y,x)] = 0. \tag{16}$$

Inspired by Dai & Wipf (2018), the proof argument involves two steps. First, we show the convergence of the parameterized density to the distribution of the ground truth measure. Then, we show that the ELBO is asymptotically tight as $\sigma \to 0$ as the KL divergence between the approximate posterior and the ground truth posterior vanishes. In particular, we do not assume that $\boldsymbol{y}$ follows a low-dimensional simple Riemann manifold with dimension less than $q$, to avoid additional definitions such as active dimension (Zheng et al., 2022). The detailed proof of Proposition A.2 is provided in Appendix

The proposition A.2 states that $\sigma$-CVAE asymptotically approximates $p_{gt}(\boldsymbol{y}|\boldsymbol{x})$ and the intractable posterior $p_\theta(\boldsymbol{z}|\boldsymbol{y},\boldsymbol{x})$ as $\sigma \to 0$ simultaneously. This proposition establishes the asymptotic expressiveness of the parameterized density $p_{\theta,\sigma}(y|x)$ as $\sigma \to 0$. The global optimality is evident from the fact that the lower bound of Eq.3 is reached. However, this proposition is limited because it does not explain the practical success of experiments in which a nonzero $\sigma$ is learned. Indeed, Proposition A.2 cannot answer why a *nonzero $\sigma$* is learned. It is not clear whether this is due to the limited capacity of the encoder/decoder network or the identifiability of the VAE.

# B Proof of proposition A.2

Following the line of proof of theorem 2 in Dai & Wipf (2018), we first consider the case when the latent dimension $d$ equals the response dimension $q$.

Step 1: Show that $\mathrm{KL}[p_{\theta_\sigma^*}(\boldsymbol{y}|\boldsymbol{x})\|p_{gt}(\boldsymbol{y}|\boldsymbol{x})] \to 0$ as $\sigma \to 0$.

Define the cumulative density function $F : \mathbb{R}^q \to [0,1]^q$ for $Y$ given $X = x$ as follows,

$$F_x(\boldsymbol{y}) = [F_1(y_1), F_2(y_2; y_1)...., F_q(y_q; y_1, ..., y_{q-1})]^T, \tag{17}$$

$$F_i(y_i; y_1, ..., y_{i-1}) = \int_{y_i'=-\infty}^{y_i} p_{gt}(y_i'|x, y_1, ..., y_{i-1})dy_i', \tag{18}$$

where $p_{gt}(y_i'|x, y_1, ..., y_{i-1})$ is the marginal distribution of the $i$-th dimension of $Y|X$, conditioned on the first $i-1$ dimensions.

By definition, we have $dF_x(\boldsymbol{y}) = p_{gt}(\boldsymbol{y}|\boldsymbol{x})d\boldsymbol{y}$. And since $p_{gt}$ is nonzero everywhere, the conditional distribution function $F_x$ is invertible. Denote its inverse by $F_x^{-1}$.

Similarly, we can define another differential and invertible function $T : \mathbb{R}^d \to [0,1]^d$

$$T(\boldsymbol{z}) = [\Phi(z_1), \Phi(z_1), ...\Phi(z_d)]^T, \tag{19}$$

where $\Phi()$ is the cumulative density function of the standard Gaussian.

Since $d = q$, we consider a possibly non-identifiable decoder parameter $\theta^* \in \{\theta | \mu_\theta(x, z) = F_x^{-1} \circ T(\boldsymbol{z}))\}$; then the corresponding density is

$$
\begin{aligned}
p_\theta(\boldsymbol{y}|\boldsymbol{x}) &= \int_{\mathbb{R}^d} \mathcal{N}(\boldsymbol{y}|\mu_\theta(\boldsymbol{x}, \boldsymbol{z}), \sigma I_d) p(\boldsymbol{z}) d\boldsymbol{z} \\
&= \int_{[0,1]^q} \mathcal{N}(\boldsymbol{y}|F_{\boldsymbol{x}}^{-1}(u), \sigma I_q) du \\
&= \int_{\mathbb{R}^q} \mathcal{N}(\boldsymbol{y}|\boldsymbol{y}', \sigma I_q) p_{gt}(\boldsymbol{y}'|\boldsymbol{x}) d\boldsymbol{y}' \to \int_{\mathbb{R}^q} \delta(\boldsymbol{y} - \boldsymbol{y}') p_{gt}(\boldsymbol{y}'|\boldsymbol{x}) d\boldsymbol{y}' = p_{gt}(\boldsymbol{y}|\boldsymbol{x}),
\end{aligned}
\tag{20}
$$

where the p.d.f of $N(\boldsymbol{y}|\boldsymbol{y}', \sigma I_q) \to \delta(\boldsymbol{y}, \boldsymbol{y}')$ when $\sigma \to 0$.

Thus, it follows immediately that $\mathrm{KL}[p_\theta(\boldsymbol{y}|\boldsymbol{x})||p_{gt}(\boldsymbol{y}|\boldsymbol{x})] \to 0$.

Step 2: Show $\forall \theta_\sigma^*, \exists \phi_\sigma^*$ s.t. $\frac{q_{\phi_\sigma^*}(\boldsymbol{z}|\boldsymbol{y},\boldsymbol{x})}{p_{\theta_\sigma^*}(\boldsymbol{z}|\boldsymbol{y},\boldsymbol{x})} \to const.$

let weights of encoder network $\phi$ are such that

$$
\begin{aligned}
\mu_\phi(\boldsymbol{x}, \boldsymbol{y}) &:= T^{-1} \circ F_{\boldsymbol{x}}(\boldsymbol{y}), \\
\Sigma_\phi(\boldsymbol{x}, \boldsymbol{y}) &:= \sigma(S_{\theta,\phi}(\boldsymbol{x}, \boldsymbol{y})^\top S_{\theta,\phi}(\boldsymbol{x}, \boldsymbol{y}))^{-1},
\end{aligned}
\tag{21}
$$

where the $d \times q$ Jacobian matrix $S_{\theta,\phi}(\boldsymbol{x}, \boldsymbol{y}) := \nabla_{\boldsymbol{z}} \mu_\theta(\boldsymbol{x}, \boldsymbol{z})|_{\boldsymbol{z}=\mu_\phi(\boldsymbol{x},\boldsymbol{y})}$.

Denote $\mu_\phi(\boldsymbol{x}, \boldsymbol{y}) = \mu_\phi, \Sigma_\phi(\boldsymbol{x}, \boldsymbol{y}) = \sigma \tilde{\Sigma}_\phi$,

$$
p_\theta(\boldsymbol{z}|\boldsymbol{y}, \boldsymbol{x}) = \frac{p_\theta(\boldsymbol{y}|\boldsymbol{x}, \boldsymbol{z})p(\boldsymbol{z})}{p_\theta(\boldsymbol{y}|\boldsymbol{x})} = \frac{\mathcal{N}(\boldsymbol{y}; \mu_\theta(\boldsymbol{x}, \boldsymbol{z}), \sigma I_m)\mathcal{N}(\boldsymbol{z}; 0, I)}{p_\theta(\boldsymbol{y}|\boldsymbol{x})}.
\tag{22}
$$

Let $\boldsymbol{z}' = (\boldsymbol{z} - \mu_\phi)/\sqrt{\sigma}$ and $q_\phi'(\boldsymbol{z}'|\boldsymbol{y}, \boldsymbol{x}), p_\theta'(\boldsymbol{z}'|\boldsymbol{y}, \boldsymbol{x})$ be the transformed pdf, then we have

$$
\begin{aligned}
\frac{q_\phi'(\boldsymbol{z}'|\boldsymbol{y}, \boldsymbol{x})}{p_\theta'(\boldsymbol{z}'|\boldsymbol{y}, \boldsymbol{x})} &= \frac{\mathcal{N}(\sqrt{\sigma}\boldsymbol{z}' + \mu_\phi; \mu_\phi, \sigma \tilde{\Sigma}_\phi) p_\theta(\boldsymbol{y}|\boldsymbol{x})}{\mathcal{N}(\boldsymbol{y}; \mu_\theta(\boldsymbol{x}, \sqrt{\sigma}\boldsymbol{z}' + \mu_\phi), \sigma I_m)\mathcal{N}(\sqrt{\sigma}\boldsymbol{z}' + \mu_\phi; 0, I)} \\
&= C(\boldsymbol{x}, \boldsymbol{y}) \exp\{-\frac{\boldsymbol{z}'^\top \tilde{\Sigma}_\phi^{-1} \boldsymbol{z}'}{2} + \frac{\|\mu_\phi + \sqrt{\sigma}\boldsymbol{z}'\|_2^2}{2} + \frac{\|\boldsymbol{y} - \mu_\theta(\boldsymbol{x}, \mu_\phi + \sqrt{\sigma}\boldsymbol{z}')\|_2^2}{2\sigma}\} \\
&= C(\boldsymbol{x}, \boldsymbol{y}) \exp\{-\frac{\boldsymbol{z}'^\top \tilde{\Sigma}_\phi^{-1} \boldsymbol{z}'}{2} + \frac{\|\mu_\phi + \sqrt{\sigma}\boldsymbol{z}'\|_2^2}{2} + \frac{\|\mu_\theta(\boldsymbol{x}, \mu_\phi) - \mu_\theta(\boldsymbol{x}, \mu_\phi + \sqrt{\sigma}\boldsymbol{z}')\|_2^2}{2\sigma}\} \\
&= C(\boldsymbol{x}, \boldsymbol{y}) \exp\{-\frac{\boldsymbol{z}'^\top \tilde{\Sigma}_\phi^{-1} \boldsymbol{z}'}{2} + \frac{\|\mu_\phi + \sqrt{\sigma}\boldsymbol{z}'\|_2^2}{2} + \frac{\|\nabla_{\boldsymbol{z}}\mu_\theta(\boldsymbol{x}, \mu_\phi)\sqrt{\sigma}\boldsymbol{z}'\|_2^2}{2\sigma}\}(\text{Taylor Expansion}) \\
&= C(\boldsymbol{x}, \boldsymbol{y}) \exp\{-\frac{\boldsymbol{z}'^\top \tilde{\Sigma}_\phi^{-1} \boldsymbol{z}'}{2} + \frac{\|\mu_\phi + \sqrt{\sigma}\boldsymbol{z}'\|_2^2}{2} + \frac{\boldsymbol{z}'^\top \nabla_{\boldsymbol{z}}\mu_\theta(\boldsymbol{x}, \mu_\phi)^\top \nabla_{\boldsymbol{z}}\mu_\theta(\boldsymbol{x}, \mu_\phi)\boldsymbol{z}'}{2}\} \\
&= C(\boldsymbol{x}, \boldsymbol{y}) \exp\{\frac{\|\mu_\phi + \sqrt{\sigma}\boldsymbol{z}'\|_2^2}{2}\} \to C(\boldsymbol{x}, \boldsymbol{y}) \exp\{\frac{\|\mu_\phi\|_2^2}{2}\} = 1.
\end{aligned}
\tag{23}
$$

Therefore, without the transformation of $\boldsymbol{z}$, $\lim_{\sigma \to 0} \mathrm{KL}[q_{\phi_\sigma^*}(\boldsymbol{z}|\boldsymbol{y}, \boldsymbol{x})||p_{\theta_\sigma^*}(\boldsymbol{z}|\boldsymbol{y}, \boldsymbol{x})] = 0$

For the case where the latent dimension $d$ is larger than the response dimension $q$, we use the first q latent dimensions to build a projection between $\boldsymbol{z}_{1:q}$ and $\boldsymbol{y}|\boldsymbol{x}$ without the remaining $d - q$ latent dimensions. To be specific, let $\mu_\theta(\boldsymbol{x}, \boldsymbol{z}) := F_x^{-1} \circ T(\boldsymbol{z}_{1:q})$, and using the same derivation of Eq.20, we get

$$
\mathrm{KL}[p_\theta(\boldsymbol{y}|\boldsymbol{x})||p_{gt}(\boldsymbol{y}|\boldsymbol{x})] \to 0.
$$

Now, we define the mean function of the encoder where $\mu_\phi(\boldsymbol{y}|\boldsymbol{x})_{1:q} = T^{-1} \circ F_{\boldsymbol{x}}(y)$ and $\mu_\phi(\boldsymbol{y}|\boldsymbol{x})_{q+1:d} = 0$ and the variance function of the encoder as follows,

$$
\Sigma_\phi(\boldsymbol{x}, \boldsymbol{y}) = \sigma[(S_{\theta,\phi}(\boldsymbol{x}, \boldsymbol{y}), \boldsymbol{n}_{q+1}, ..., \boldsymbol{n}_d)^T (S_{\theta,\phi}(\boldsymbol{x}, \boldsymbol{y}), \boldsymbol{n}_{q+1}, ..., \boldsymbol{n}_d)]^{-1},
$$

where $\boldsymbol{n}_{i=q+1}^{d}$ are a set of $q$-dimensional column vectors, e.g. an orthonormal basis of $\mathrm{null}(S_{\theta,\phi})$, such that

$$
\begin{aligned}
S_{\theta,\phi}^{T}\boldsymbol{n}_i &= 0, \\
\boldsymbol{n}_i^{T}\boldsymbol{n}_j &= \mathbf{1}_{i=j}.
\end{aligned}
\tag{24}
$$

The set of $\boldsymbol{n}_{i=q+1}^{d}$ always exists due to the fact that $S_{\theta,\phi}$ is the $d \times q$ Jacobian matrix with a null space of at least $d - q$.

This implies,

$$
\Sigma_\phi(\boldsymbol{x},\boldsymbol{y}) = \begin{bmatrix} \sigma(S_{\theta,\phi}(\boldsymbol{x},\boldsymbol{y})^{\top}S_{\theta,\phi}(\boldsymbol{x},\boldsymbol{y}))^{-1} & \mathbf{0} \\ \mathbf{0} & I_{d-q} \end{bmatrix}.
\tag{25}
$$

Thus, The first $q$ dimensions of $q_\phi(\boldsymbol{z}|\boldsymbol{y},\boldsymbol{x})$ can exactly match the first $q$ dimensions of true posterior as shown in Eq.23. The remaining $d - q$ dimensions follow a standardized Gaussian distribution that matches the posterior of $p_{\theta_\sigma^*}(\boldsymbol{z}|\boldsymbol{y},\boldsymbol{x})$ due to the fact that these dimensions are not used in likelihood involving $\mu_\theta(\boldsymbol{x},\boldsymbol{z}) := F_x^{-1} \circ T(\boldsymbol{z}_{1:q})$ as it remains to follow the prior distribution. As a result, $\lim_{\sigma \to 0} \mathrm{KL}[q_{\phi_\sigma^*}(\boldsymbol{z}|\boldsymbol{y},\boldsymbol{x})||p_{\theta_\sigma^*}(\boldsymbol{z}|\boldsymbol{y},\boldsymbol{x})] = 0$.

## C Proof of theorem 3.4

We consider $G_m^*$ as follows,

$$
G_m^* = f_0^\eta + f_{\boldsymbol{x}}^\eta(\boldsymbol{x}) + f_{\boldsymbol{z}_1}^\eta(\boldsymbol{z}_1) + f_{\boldsymbol{z}_2}^\eta(\boldsymbol{z}_2) + f_{\boldsymbol{x}\boldsymbol{z}_1}^\eta(\boldsymbol{x},\boldsymbol{z}_1) + f_{\boldsymbol{x}\boldsymbol{z}_2}^\eta(\boldsymbol{x},\boldsymbol{z}_2) + f_{\boldsymbol{z}_1\boldsymbol{z}_2}^\eta(\boldsymbol{z}_1,\boldsymbol{z}_2) + f_{\boldsymbol{x}\boldsymbol{z}_1\boldsymbol{z}_2}^\eta(\boldsymbol{x},\boldsymbol{z}_1,\boldsymbol{z}_2)
\tag{26}
$$

Step 1: we prove, under the assumption 3.3, that there exists a unique decomposition of $G_m^*$ in Eq.?? up to a rotation of coordinates within the block. Specifically, given $G_m^*$, there is a unique form for $f_0^\eta, f_{\boldsymbol{x}}^\eta(\boldsymbol{x}), ..., f_{\boldsymbol{z}_1\boldsymbol{z}_2}^\eta(\boldsymbol{z}_1,\boldsymbol{z}_2)$, and $f_{\boldsymbol{x}\boldsymbol{z}_1\boldsymbol{z}_2}^\eta(\boldsymbol{x},\boldsymbol{z}_1,\boldsymbol{z}_2)$.

Following the line of proof of Theorem 1 Soboĺ (1993), with an abuse of notation $\mu$, we can see that $f_0^\eta = \int G_m^*(\boldsymbol{x},\boldsymbol{z}_1,\boldsymbol{z}_2)\mu(d\boldsymbol{x}d\boldsymbol{z}_1d\boldsymbol{z}_2) = \mathbb{E}[Y]$, as $G_m^*$ is the transformation of $X, Z_1, Z_2$ to generate $Y$.

Then, the integration of $G_m^*(\boldsymbol{x},\boldsymbol{z}_1,\boldsymbol{z}_2)$ over any two block dimensions, namely $\int G_m^*(\boldsymbol{x},\boldsymbol{z}_1,\boldsymbol{z}_2)\mu(d\boldsymbol{z}_1d\boldsymbol{z}_2)$, admits the following,

$$
\begin{aligned}
\int G_m^*(\boldsymbol{x},\boldsymbol{z}_1,\boldsymbol{z}_2)\mu(d\boldsymbol{z}_1d\boldsymbol{z}_2) =& f_0^\eta + f_{\boldsymbol{x}}^\eta(\boldsymbol{x}) + \int \overbrace{f_{\boldsymbol{z}_1}^\eta(\boldsymbol{z}_1)\mu(d\boldsymbol{z}_1)}^{0} + \overbrace{f_{\boldsymbol{z}_2}^\eta(\boldsymbol{z}_2)\mu(d\boldsymbol{z}_2)}^{0} \\
&+ \int \overbrace{f_{\boldsymbol{x}\boldsymbol{z}_1}^\eta(\boldsymbol{x},\boldsymbol{z}_1)\mu(d\boldsymbol{z}_1)}^{0} + \overbrace{f_{\boldsymbol{x}\boldsymbol{z}_2}^\eta(\boldsymbol{x},\boldsymbol{z}_2)\mu(d\boldsymbol{z}_2)}^{0} \\
&+ \int \overbrace{f_{\boldsymbol{z}_1\boldsymbol{z}_2}^\eta(\boldsymbol{z}_1,\boldsymbol{z}_2)\mu(d\boldsymbol{z}_1d\boldsymbol{z}_2)}^{0} + \int \overbrace{f_{\boldsymbol{x}\boldsymbol{z}_1\boldsymbol{z}_2}^\eta(\boldsymbol{x},\boldsymbol{z}_1,\boldsymbol{z}_2)\mu(d\boldsymbol{z}_1d\boldsymbol{z}_2)}^{0},
\end{aligned}
\tag{27}
$$

where the cancellation to zeros is a direct result of assumption 3.3.

Therefore,

$$
\begin{aligned}
f_{\boldsymbol{x}}^\eta(\boldsymbol{x}) &:= \int G_m^*(\boldsymbol{x},\boldsymbol{z}_1,\boldsymbol{z}_2)\mu(d\boldsymbol{z}_1d\boldsymbol{z}_2) - f_0^\eta, \\
f_{\boldsymbol{z}_1}^\eta(\boldsymbol{z}_1) &:= \int G_m^*(\boldsymbol{x},\boldsymbol{z}_1,\boldsymbol{z}_2)\mu(d\boldsymbol{x}d\boldsymbol{z}_2) - f_0^\eta, \\
f_{\boldsymbol{z}_2}^\eta(\boldsymbol{z}_2) &:= \int G_m^*(\boldsymbol{x},\boldsymbol{z}_1,\boldsymbol{z}_2)\mu(d\boldsymbol{x}d\boldsymbol{z}_1) - f_0^\eta.
\end{aligned}
\tag{28}
$$

Using the same argument, we can see

$$f^\eta_{\boldsymbol{xz}_1}(\boldsymbol{x}, \boldsymbol{z}_1) := \int G^*_m(\boldsymbol{x}, \boldsymbol{z}_1, \boldsymbol{z}_2)\mu(dz_2) - \int G^*_m(\boldsymbol{x}, \boldsymbol{z}_1, \boldsymbol{z}_2)\mu(d\boldsymbol{x}dz_2) - \int G^*_m(\boldsymbol{x}, \boldsymbol{z}_1, \boldsymbol{z}_2)\mu(dz_1dz_2) + f^\eta_0,$$

$$f^\eta_{\boldsymbol{xz}_2}(\boldsymbol{x}, \boldsymbol{z}_2) := \int G^*_m(\boldsymbol{x}, \boldsymbol{z}_1, \boldsymbol{z}_2)\mu(dz_1) - \int G^*_m(\boldsymbol{x}, \boldsymbol{z}_1, \boldsymbol{z}_2)\mu(d\boldsymbol{x}dz_1) - \int G^*_m(\boldsymbol{x}, \boldsymbol{z}_1, \boldsymbol{z}_2)\mu(dz_1dz_2) + f^\eta_0,$$

$$f^\eta_{\boldsymbol{xz}_2}(\boldsymbol{z}_1, \boldsymbol{z}_2) := \int G^*_m(\boldsymbol{x}, \boldsymbol{z}_1, \boldsymbol{z}_2)\mu(d\boldsymbol{x}) - \int G^*_m(\boldsymbol{x}, \boldsymbol{z}_1, \boldsymbol{z}_2)\mu(d\boldsymbol{x}dz_2) - \int G^*_m(\boldsymbol{x}, \boldsymbol{z}_1, \boldsymbol{z}_2)\mu(d\boldsymbol{x}dz_1) + f^\eta_0,$$

$$\tag{29}$$

and lastly, the residual

$$f^\eta_{\boldsymbol{xz}_1\boldsymbol{z}_2}(\boldsymbol{x}, \boldsymbol{z}_1, \boldsymbol{z}_2) := G^*_m(\boldsymbol{x}, \boldsymbol{z}_1, \boldsymbol{z}_2) - (f^\eta_{\boldsymbol{x}}(\boldsymbol{x}) + f^\eta_{\boldsymbol{z}_1}(\boldsymbol{z}_1) + f^\eta_{\boldsymbol{z}_2}(\boldsymbol{z}_2) + f^\eta_{\boldsymbol{xz}_1}(\boldsymbol{x}, \boldsymbol{z}_1) + f^\eta_{\boldsymbol{xz}_2}(\boldsymbol{x}, \boldsymbol{z}_2) + f^\eta_{\boldsymbol{z}_1\boldsymbol{z}_2}(\boldsymbol{z}_1, \boldsymbol{z}_2) + f^\eta_0).$$

$$\tag{30}$$

This shows that all $f^\eta$ are unique given $G^*_m$

Step 2: We prove that, with additional assumptions including $f^\eta_{\boldsymbol{z}_2}(\boldsymbol{z}_2) = \sigma^* \boldsymbol{z}_2$, $\sigma$-CVAE recovers the ground truth $G^*_m$.

By comparison of block coordinates, $\mu_\sigma(\boldsymbol{x}, \boldsymbol{z}_1) + \sigma \boldsymbol{z}_2 = G^*_m$ is almost surely true if the following holds for all $\boldsymbol{x}, \boldsymbol{z}_1, \boldsymbol{z}_2$ (up to a zero measure of $\boldsymbol{z}_2$).

$$\mu_\theta(\boldsymbol{x}, \boldsymbol{z}_1) = f^\eta_{\boldsymbol{x}}(\boldsymbol{x}) + f^\eta_{\boldsymbol{z}_1}(\boldsymbol{z}_1) + f^\eta_{\boldsymbol{xz}_1}(\boldsymbol{x}, \boldsymbol{z}_1) + f^\eta_0 + C_0$$
$$\sigma \boldsymbol{z}_2 = f^\eta_{\boldsymbol{z}_2}(\boldsymbol{z}_2) \tag{31}$$
$$f^\eta_{\boldsymbol{xz}_2}(\boldsymbol{z}_1, \boldsymbol{z}_2) + f^\eta_{\boldsymbol{xz}_2}(\boldsymbol{x}, \boldsymbol{z}_2) + f_{\boldsymbol{xz}_1\boldsymbol{z}_2}(\boldsymbol{x}, \boldsymbol{z}_1, \boldsymbol{z}_2) = C_0$$

Where $C_0$ is a constant independent of $x, z_1, z_2$.

In alignment with the identifiability assumption 3.3, $f^\eta_{\boldsymbol{z}_2}(\boldsymbol{z}_2)$ cannot have any additive constant. Specifically, $\int f^\eta_{\boldsymbol{z}_2}(\boldsymbol{z}_2)\mu(\boldsymbol{z}_2) = \int \sigma \boldsymbol{z}_2\mu(\boldsymbol{z}_2) = 0$. Assuming the universal approximation theorem of $\mu_\theta(\boldsymbol{x}, \boldsymbol{z}_1)$, the results are immediately implied by considering the additional assumption. The conditions and proof of the universal approximation property can be found in Hornik et al. (1989)

## D   Comparison between Theorem 3.4 and existing approximation theories.

Table 5: Comparison between Theorem 3.4 and existing approximation theories.

| Approximation theory | Key Limitation on oracle $\theta, \sigma$ | Existence of optimal Gaussian encoders | Enough to explain a good fit of two-moon data? |
|---|---|---|---|
| Proposition A.2 | $\sigma \to 0$ | Yes. Asymptotically $\sigma \to 0$ | No. The learned decoder recovers the two-moon data generative process without taking $\sigma \to 0$. |
| Linear VAE Lucas et al. (2019b) | $\theta$ represents a linear function between $\mu$ and $z$. | Yes. Closed-form Gaussian posterior within the mean-field Gaussian encoder family. | No. Two-moon data is has non-linear mean function w.r.t Z at $x = -1, 1$. |
| Theorem 3.4 | No additional limitation. | No. It is not guaranteed. | Yes. The learned decoder $\sigma$ recovers the data-generating variance. |

# E    An toy example of inflated estimate of $\sigma^2$ due to suboptimal inference (via Fisher's identity)

This serves as a solid example of how a less concentrated $q_\phi$ leads to inflated $\sigma$. For simplicity, we consider the 1D unconditional case, which in statistics, is called the many-normal-means problem.

We are given $\boldsymbol{x}$ sample from the following $p(\boldsymbol{z}) \sim N(0,1), p(\boldsymbol{x}|\boldsymbol{z}) \sim N(\boldsymbol{z}, \sigma^2)$.

Easily, we have the following:

$$p(\boldsymbol{x}) \sim N(0, \sigma^2 + 1), p(\boldsymbol{z}|\boldsymbol{x}) \sim N(\frac{x}{1+\sigma^2}, \frac{\sigma^2}{1+\sigma^2}).$$

Using Fisher's identity, the gradient of the marginal log-likelihood $p(\boldsymbol{x})$ given $\boldsymbol{x}_i$

$$\frac{\partial \log p(\boldsymbol{x})}{\partial \sigma^2} = \mathbb{E}_{p(\boldsymbol{z}|\boldsymbol{x})}[-\frac{1}{2\sigma^2} + \frac{(\boldsymbol{x}-\boldsymbol{z})^2}{2\sigma^4}] = -\frac{1}{2\sigma^2} + \frac{1}{2\sigma^4}\mathbb{E}_{p(\boldsymbol{z}|\boldsymbol{x})}[(\boldsymbol{x}-\boldsymbol{z})^2],$$

, where $\mathbb{E}_{p(\boldsymbol{z}|\boldsymbol{x})}[(x-z)^2] = (x-\mu_z)^2 + \sigma_z^2$, and $\mu_z := \frac{x}{1+\sigma^2}, \sigma_z := \frac{\sigma^2}{1+\sigma^2}$

By setting $\mathbb{E}_{p(\boldsymbol{z}|\boldsymbol{x})}[\frac{\partial \log p(\boldsymbol{x}|\boldsymbol{z})}{\partial \sigma^2}] = 0$, we have $\hat{\sigma}^2_{mle} = x^2 - 1$, which corresponds to MLE of $\frac{\partial \log p(\boldsymbol{x})}{\partial \sigma^2} = 0$

Now if $q(\boldsymbol{z}|\boldsymbol{x})$ is less concentrated than $p(\boldsymbol{z}|\boldsymbol{x})$, e.g.,

$$q(\boldsymbol{z}|\boldsymbol{x}) \sim N(\mu_z, \sigma_q^2)$$

where $\sigma_q^2 = k\sigma_z^2$ for some $k > 1$.

$$
\begin{aligned}
\frac{\partial ELBO}{\partial \sigma^2} &= \mathbb{E}_{q(\boldsymbol{z}|\boldsymbol{x})}[-\frac{1}{2\sigma^2} + \frac{(x-z)^2}{2\sigma^4}] = -\frac{1}{2\sigma^2} + \frac{1}{2\sigma^4}[(x-\mu_z)^2 + k\sigma_z^2] \\
&= -\frac{1}{2\sigma^2} + \frac{1}{2\sigma^4}(\frac{\sigma^4 x^2}{(1+\sigma^2)^2} + \frac{k\sigma^2}{(1+\sigma^2)}) \\
&= \frac{\sigma^2 x^2 + k(1+\sigma^2) - (1+\sigma^2)^2}{2\sigma^2(1+\sigma^2)^2} \\
&\propto -\sigma^4 + (x^2 + k - 2)\sigma^2 + k - 1
\end{aligned}
$$

Notice that the only solution of $\partial ELBO/\partial \sigma^2 = 0$ is

$$\hat{\sigma}^2_{eb} = \frac{x^2 + k - 2 + \sqrt{(x^2 + k - 2)^2 + 4(k-1)}}{2} > x^2 + k - 2 > x^2 - 1 = \sigma^2_{mle}$$

Thus, such $q_\phi$ introduces a positive bias to the MLE of $\sigma$ in log-likelihood, but it reduces the possibility of an ill-defined likelihood. For example, $|x| \leq 1$ leads to invalid estimates of $\sigma^2_{mle}$, and thus James-Stein shrinkage (JAMES, 1961) may be considered; it can be avoided by $q_\phi$ if $k > 2$. In other words, we suffer from the numerical instability of suboptimal $q_\phi$, rather than likelihood blowup on a possibly near-zero $\sigma$. Nonetheless, the estimate of $\sigma$ from a less concentrated $q_\phi$ is inflated.

# F    Experiment and algorithm details

We implemented the proposed method using PyTorch 1.8.2 +cu111 with Python 3.7 on an Ubuntu internal cluster with multiple Nvidia GPUs including A10,A30, A100, A100-40GB, A100-80GB, and V100. We are not aware of which GPU is used in the experiments due to the task distribution service.

In Section 5.1, we used fully connected 4-layer neural networks with a hyperbolic tangent activation function for the encoding and decoding networks. The latent dimension is set to 2, and the width of the hidden layers

is $[16, 8, 4, 2]$ and $[2, 4, 16, 4]$, respectively. $\sigma$ initialized at 1. The batch size is equal to the sample size of the training data. We generate 5000 data points with $2,500$ for each class, with a latent variable generated from a standard normal distribution.

In Section 5.2, we used 3 simulation datasets from the following:

$M_1$ is a non-linear model with additive Gaussian noise:

$$\boldsymbol{y} = \boldsymbol{x}_1^2 + \exp\left(\boldsymbol{x}_2 + \boldsymbol{x}_3/3\right) + \sin\left(\boldsymbol{x}_4 + \boldsymbol{x}_5\right) + \varepsilon, \text{ where } \varepsilon \sim N(0, 1).$$

$M_2$ is A nonlinear model with multiplicative non-Gaussian noise:

$$\boldsymbol{y} = \left(5 + \boldsymbol{x}_1^2/3 + \boldsymbol{x}_2^2 + \boldsymbol{x}_3^2 + \boldsymbol{x}_4 + \boldsymbol{x}_5\right) * \exp(0.5 \times \varepsilon), \text{where, } \varepsilon \sim 0.5N(-2, 1) + 0.5N(2, 1).$$

$M_3$. A Gaussian Mixture Model:

$$\boldsymbol{y} \sim \begin{cases} N(-1 - \boldsymbol{x}_1 - 0.5\boldsymbol{x}_2, 0.5^2), & \text{if } U = 0, \\ N(1 + \boldsymbol{x}_1 + 0.5\boldsymbol{x}_2, 1^2), & \text{if } U = 1, \end{cases}, \text{ where } U \sim \text{Binomial}(1, 0.7).$$

The predictor dimension is 5 for $M_1$ and $M_2$, and 2 for $M_3$. The response $\boldsymbol{y}$ is univariate. The sample size $N = 5000$, the test sample size is $k = 2000$. The mean squared error of the mean is $\frac{1}{k}\sum_{i=1}^{k}\left(\hat{\mathbb{E}}[\boldsymbol{y}|\boldsymbol{x}_k] - \mathbb{E}[\boldsymbol{y}|\boldsymbol{x}_k]\right)^2$. The mean squared error of the standard deviation is $\frac{1}{k}\sum_{i=1}^{k}\left(\hat{\text{SD}}[\boldsymbol{y}|\boldsymbol{x}_k] - \text{SD}[\boldsymbol{y}|\boldsymbol{x}_k]\right)^2$.

We used fully connected 5-layer neural networks with a hyperbolic tangent activation function for the encoding and decoding network. The latent dimension is set to 5. The width of the hidden layer of the network is $[32, 16, 8, 4]$, and $[8, 32, 16, 4]$. For the WGCS method, the conditional generator G is parameterized using a fully connected neural network. The discriminator D is parameterized using a fully connected two-layer neural network. The noise vector is $\eta \sim N(0, 1)$. For the NNKCDE method, the tuning parameters are chosen by cross-validation. The bandwidth of CKDE is determined based on the standard formula $h = 1.06\sigma n^{-1/(2t+d)}$, where $\sigma$ is a measure of spread, $t$ is the order of the kernel, and $d$ is the dimension of $X$. The FlexCode basis expansion-based method uses the Fourier basis. The maximum number of bases is set to 40, and the actual number of bases is selected using cross-validation. WGCS generates $10,000$ Monte Carlo samples to estimate the conditional distribution of each test $x_k$ to calculate the conditional mean and conditional standard deviation, while our method uses only 500 samples. For other methods, the estimates are calculated by numerical integration.

# G    More discussion for experiments in Section 5.1

Additional experiment of learned $\sigma$-CVAE using Two moon dataset with $\tau = 0.1$. The ground truth $\sigma$ recovers $\tau$, which is 0.1.

**Additional experiments on wall time until convergence.** In Table 6, under different $K$ epochs for updating $\theta, \phi$, we report the largest wall time out of 5 repetitions of successful experiments with the estimated $\sigma$ initialized at 1, converging to the true value of 0.1. Tolerance $C = 0.05$. The same criterion for convergence is used in each $K$. NA represents no correct estimate of the true value $\sigma$ within a 3 % relative error. We find that the success rate of experiments for uncalibrated $\sigma$-CVAE increases with $K$ and due to calibration in the sufficiently large $K$, our proposed algorithm might be slower than an uncalibrated one. Time is measured by the Python function time.perf_counter().

Table 6: Comparison of wall time for convergence

| Inner step size K | 25 | 50 | 100 | 250 | 500 |
|---|---|---|---|---|---|
| Uncalibrated | NA | NA | 62.15 | 76.10 | **177.81** |
| Calibrated w./ C=0.05 | **10.55** | **12.69** | **25.00** | **33.72** | 269.78 |

**Calibration as warm restart.** Nonetheless, the calibration step in our framework is beneficial for the estimation of $\sigma$-CVAE even if *optimal* $\sigma$ is learned. As shown by Lucas et al. (2019b), posterior collapse in $q_\phi$ occurs even if optimal $\sigma$ is learned, possibly due to local optima in $q_\phi$. Therefore, when calibrating $\sigma$,

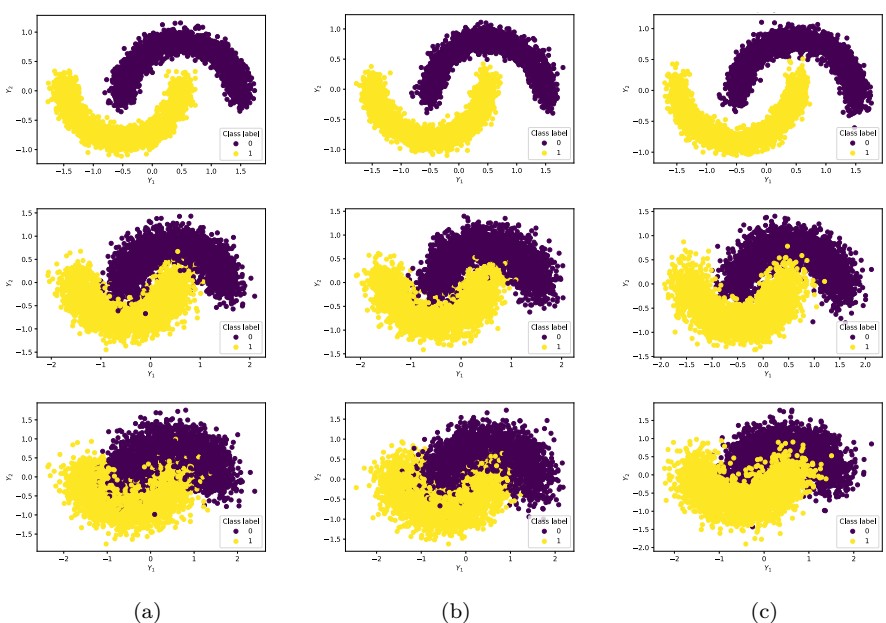

Figure 4: Calibrated $\sigma$-CVAE on two moon dataset: (a) the training data of size 5000, (b) estimated training data, and (c) sample drawn from calibrated $\sigma$-CVAE. The true *sigma*, $\tau = 0.1, 0.2$ and $0.3$ from top to bottom. Best viewed in color.

the training dynamics of the parameter $\phi$ leverage the rapid change of $\sigma$, perhaps due to a smoother loss landscape with larger $\sigma$ Dai et al. (2021), taking advantage to escape the local optima and converge to a possibly better local optimum at the time when $\sigma$ returns to optimally small again. Even if we calibrate the $\sigma$ back to initialization, the optimization that was done before this calibration would be beneficial as a warm restart. To this extent, our calibration step is an inexpensive and implicit KL annealing scheme without a predefined monotonic or cyclic schedule Skafte et al. (2019); Stirn & Knowles (2020). Therefore, we recommend that this calibration step should be used even if a sufficiently large $K$ is used. Please note that warm restart does not guarantee the convergence of the algorithm in the non-convex ELBO optimization; throughout the experiments, we observe that cold restart is necessary for instances with very poor initialization.

### G.1 More discussion with existing KL annealing methods.

The explicit KL annealing is undoubtedly easy and simple to implement. However, it is less efficient and often requires more computational power in terms of preventing posterior collapse. For example, if $\sigma$ is assumed to be fixed, an explicit KL annealing schedule like Eq.12 is essentially a series of deterministic $\sigma$ calibrations in Eq.13, due to $\sigma$-$\beta$ equivalence. Therefore, the annealing process of the loss landscape is predefined and not adaptive to the performance of encoders. Similarly, when $\sigma$ is a learned parameter, an explicit KL annealing schedule confounds with it in shaping the loss landscape, similarly to how it would by a single "effective" $\sigma_e$. For example, when $\beta = 0.25$ is added to the LHS of Eq.4, the "effective" $\sigma_e$ is half of the current $\sigma$ value, i.e. $\sigma_e = 0.5 * \sigma$ . The confounding between $\beta$ and $\sigma$ will not vanish unless $\beta$ is 1. In extreme cases, if the predefined $\beta_t = 1/(2\sigma_t^2)$, the effective $\sigma_e$ would be fixed at 1. This means that there would be no KL annealing to the loss landscape at all because there would be no change in the effective $\sigma_e$. Such confounding not only complicates the annealing process of the loss landscape determined by the effective $\sigma_e$, but also hinders the accurate and efficient estimation of $\sigma$ under the maximum likelihood principle.

Experiment details of Table 1 & 2. We trained the Gaussian $\sigma$-CVAE with a learned $\sigma$ using a monotonic or a cyclical annealing schedule, tuning $\beta$ linearly from 0 to 1. For hyperparameters in monotonic annealing, we use a cycle period $M = 10$ of 5 maximum cycles with a balanced ratio $R = 0.5$. In their original notation,

the number of cycles is 5, the ratio is 0.5, and the total iterations is 50. For hyperparameters in monotonic annealing, we consider a single monotonic annealing with the same length $M*5 = 50$. The $\beta$ becomes a fixed value of 1 after a maximum of 50 iterations is reached until the algorithm converges. The hyperparameters of KL annealing are chosen reasonably for a fair comparison. The maximum wall time until convergence is recorded by the Python function *time.perf_counter()* out of 5 successful repetitions.

## H    Ablation experiments on the effect of calibration

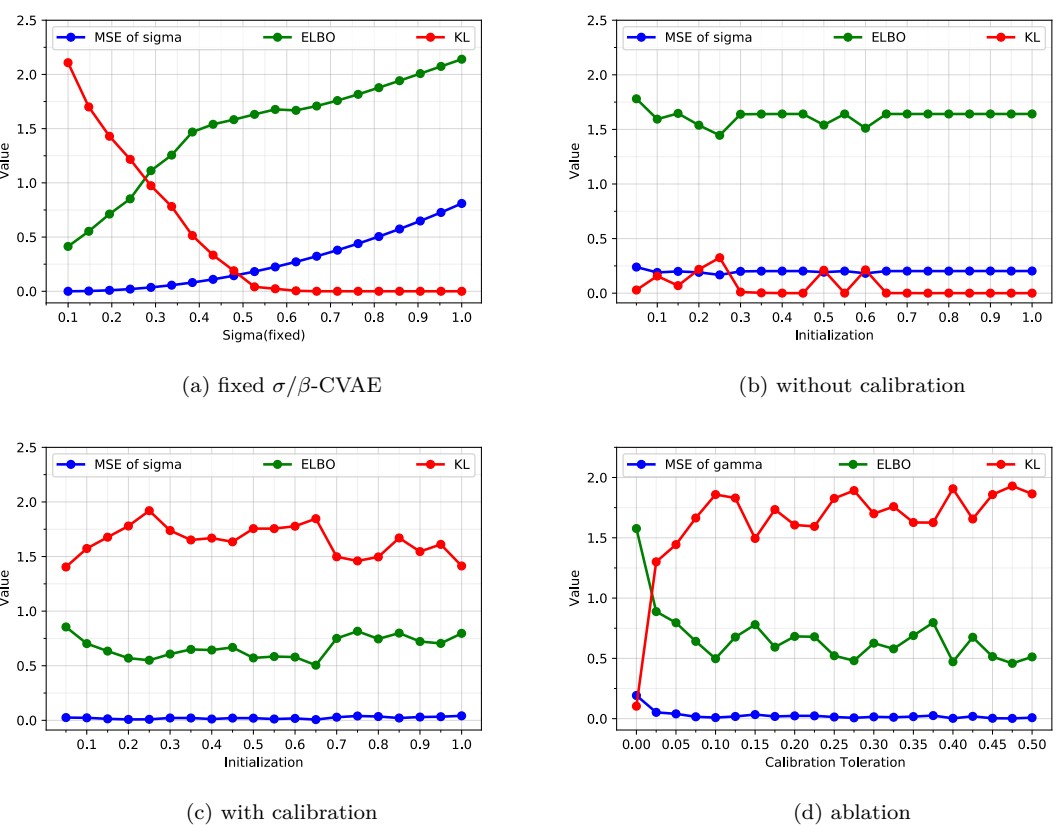

(a) fixed $\sigma/\beta$-CVAE

(b) without calibration

(c) with calibration

(d) ablation

Figure 5: Experiments on twomoon dataset with ground truth $\sigma = 0.1$. Results of the CVAE models are reported using (a) a fixed $\sigma$, equivalent to $\beta$-CVAE with a predefined $\beta$ (b) a learned $\sigma$ without calibration with tolerance 0.05, (c) a learned sigma with calibration, and (d) a learned sigma with difference toleration for calibration which is intialized at 1. $K$ is set to 25. Each data point is averaged over 20 repetitions. MSE of sigma is calculated by the squared error between the learned/fixed sigma and the ground truth.

In this section, more experiments are performed to highlight the importance of calibration using the two-moon dataset in Section 5.1, a nontrivial dataset simulated from a nonlinear generating function that causes CVAE models to have a reproducible posterior collapse problem. Throughout this section, we use $\sigma = 0.1$ to generate our training dataset once and repeat our experiment 20 times. We use small $K = 25$.

To reiterate the importance of learning $\sigma$ in CVAE models, we report the results of those using fixed $\sigma$ ranging from 0.1 to 1 in Figure 5(a). CVAE are well specified if the predefined fixed $\sigma$ is close to the ground truth value. The performance of CVAE degenerates rapidly as the difference between the fixed $\sigma$ and the ground truth increases. After the predefined $\sigma$ is larger than a certain point, the posterior collapse in CVAE is exhibited by vanishing KL divergence between the approximate posterior and prior. Therefore, we should not use CVAE with a predefined variance scalar $\sigma$ when the ground truth $\sigma$ in the data is unapproachable.

To emphasize the numerical instability in the training dynamics of $\sigma$ and the posterior collapse of the learned model, we report the results of CVAE that iteratively update $\sigma$ without calibration in Figure 5(b) with *various initializations of $\sigma$ ranging from 0.05 to 1*. The learned CVAE model ends up with a very high ELBO, fails to recover the ground truth $\sigma$, and the KL divergence between the approximate posterior and prior vanishes in most cases regardless of the initialization of $\sigma$. In Figure 1, we can see that $\sigma$ is trapped around 0.5 if no calibration is provided. It is known that the non-convex landscape of the ELBO function w.r.t. $\theta, \phi$ leads to many local optima, but our experiment specifically reveals the difficulty in estimating $\sigma$ in a dual-step algorithm.

To provide more empirical evidence of our proposed method, we report the results of our calibrated method in Figure 5(c) for different initializations of $\sigma$ ranging from 0.05 to 1. The tolerance that triggers our proposed calibration step is set to 0.05. The results show the consistency of our proposed method in accurately estimating the truth of the ground $\sigma$, preventing posterior collapse, and thus obtaining locally optimal CVAE models.

To ablate the calibration effect, we report the results of our proposed method with various tolerance hyperparameters $C$ ranging from 0.00 to 0.5 in Figure 5(d). We initialized $\sigma$ at 1. In the case that the tolerance is 0, our proposed calibration step will never be triggered due to the non-negativity of KL divergence. We observe that a small tolerance of 0.05 is empirically effective in ameliorating posterior collapse and robust estimation of $\sigma$. Note that the tolerance setting is case-by-case and may not be generalized to other datasets, and a prodigious tolerance, which goes beyond the possible value of KL divergence, leads to non-convergence in the algorithm. Therefore, we recommend a relatively small tolerance hyperparameter that triggers fewer calibrations and offers less extra computational burden.

## I    Image reconstruction: MNIST

MNIST dataset LeCun et al. (1998) contains $60,000$ gray images with size $28 \times 28$. Dividing each image into two parts $\boldsymbol{x}, \boldsymbol{y}$, we treat one part of the image as the predictor $\boldsymbol{x}$ and the rest of the image as the response $\boldsymbol{y}$. In the Appendix I, we use fully connected 3-layer neural networks with a ReLU activation function for the encoding and decoding network. The latent dimension is set to 5. The dimensions of $X$ are 196, 392, and 588, and the corresponding dimensions of $Y$ are 588, 392, and 196. Two fully connected hidden layers are 256 and 128 for the encoder, and we reverse the width in the decoders. The batch size is 100. We update and calibrate $\sigma$ per 100 iterations with a posterior collapse tolerance set to 0.001. In Figure 6, we consider three situations in which $1/4$, $1/2$, and $3/4$ of the image are observed, and the goal is to learn the distribution of the rest of the image. We chose 10 images from the test dataset to evaluate our method.

We see that the generated images are similar to the truth, with reasonable variations, and the variations in reconstruction decrease as a larger part of the image is observed.

## J    Image generation: CelebA

The CelebA dataset Liu et al. (2015) contains more than $200,000$ colored celebrity face images with 40 facial attribute annotations. In this section, we used the architect of the CVAE models from Hou et al. (2019) and embedded the attribute label of a vector in the images channels. If a label in one dimension is 1, we embed it as an image channel of size $96 \times 96$ that takes a value of 1 on every pixel. The width of the hidden layers is set to $[32, 64, 128, 256, 512]$ and the convolution layer with a kernel size of 3 , a stride size of 2, and a padding size of 1. We used LeakyReLU activation functions at a negative slope of 0.02 followed by a batch normalization layer. The number of latent dimensions is set to 32. The batch size is 16. We update and calibrate each gamma per 1809 iterations, which is a factor of total training sample size and posterior collapse tolerance set to 0.05. We apply the proposed method to the generation of human face images with binary label features as a predictor. Since some of the attributes are highly correlated, we validate our proposed framework using the following attributes *Male, Young, Eyeglasses, Bald, Mustache, Smiling* to be the predictor $\boldsymbol{x}$, and the response $\boldsymbol{y}$ here is the center-chopped scaled images with a size of $96 \times 96$. We show the six types of true images and the generated images in each row of Figure 7. The attribute labels

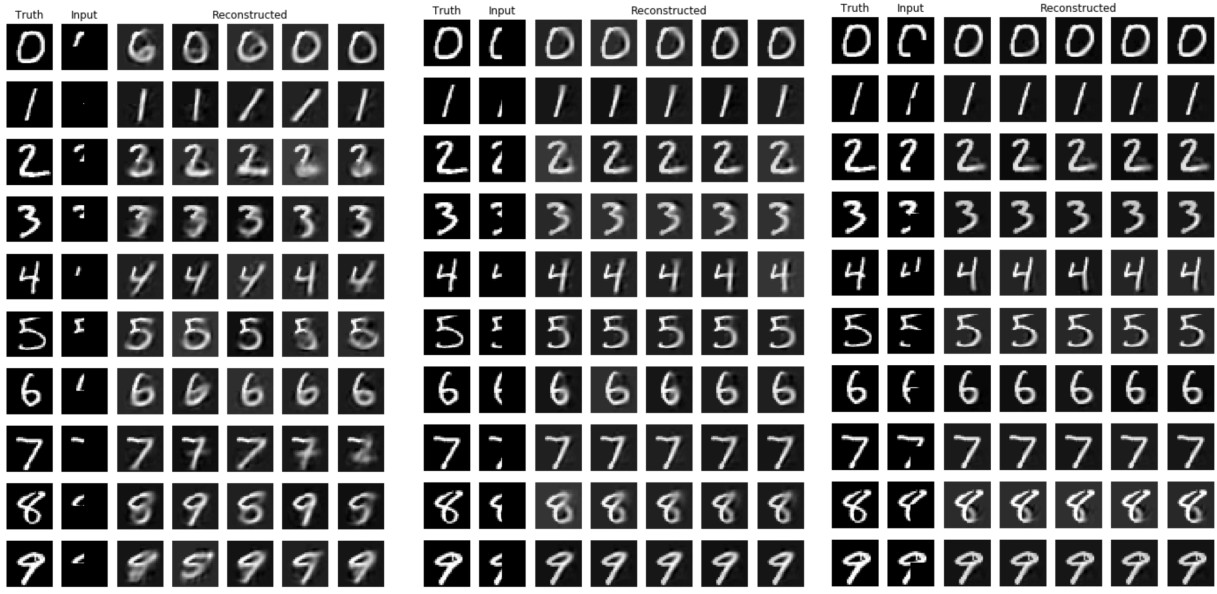

Figure 6: Reconstructing images in the MNIST test dataset. The left column of each panel contains the true images from the test dataset that are not used for training. The second left column contains the given part of the image that are used as predictor (i.e. upper left 1/4 in (a), left half 1/2 in (b), and upper left half 3/4 in (c)), and the rest 5 columns are the reconstructed images.

for these three types are shown in Table 7. Our generated images preserve the face attributes as input with moderate clarity.

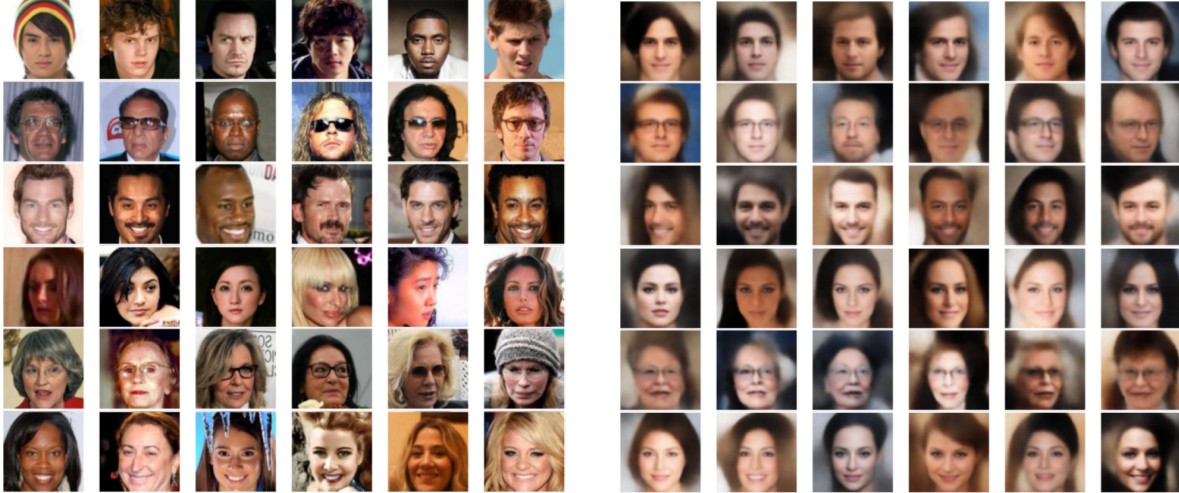

Figure 7: (a) The true images in CelebA. Images of the same row have the same attributes. (b) Generated (not reconstructed) images with the same attributes. Each row corresponds to a specific type of face identical to the same row of (a)

**Limitations** Although our simple calibration $\sigma$-CVAE method does not provide a state-of-the-art image generation framework, we hypothesize that the conditional distribution of the face image may not satisfy the assumption 3.3. Expanding the analysis in consideration of modern variants/techniques that are specific for (conditional) imagine generation is important subsequent work. Further efforts are needed to make it a state-of-the-art (C)VAE variants for conditional image generator. One common challenge in conditional image generation is meaningful quantitative measures to generated samples. Metrics such as test likelihood

or MSE failed in measuring the clarity of generated face images, and the ones like FID or Inception Score add extra mathematical assumptions that are difficult to validate.

Table 7: Attributes for six types of face images in Figure

| | MALE | YOUNG | EYEGLASSES | BALD | MUSTACHE | SMILE |
|---|---|---|---|---|---|---|
| TYPE 1 | +1 | +1 | -1 | -1 | -1 | -1 |
| TYPE 2 | +1 | -1 | +1 | -1 | -1 | -1 |
| TYPE 3 | +1 | +1 | -1 | -1 | +1 | +1 |
| TYPE 4 | -1 | +1 | -1 | -1 | -1 | -1 |
| TYPE 5 | -1 | -1 | +1 | -1 | -1 | -1 |
| TYPE 6 | -1 | +1 | -1 | -1 | -1 | +1 |

