# OpenReview forum: "Doubly Robust Conditional VAE via Decoder Calibration: An Implicit KL Annealing Approach"
_TMLR — Accepted by TMLR_

### Review · Reviewer_gj1z · 2024-10-03

**Summary Of Contributions:**

The paper aims to provide a theoretical analysis for two commonly used components in improving VAEs and their training dynamics is practice: (1) using $\sigma$ as a model parameter or hyper-parameter of the decoder when calculating the likelihood, or the “reconstruction loss”; and (2) using the $\beta$ hyper-parameter to weigh the KL-divergence term in the ELBO. The authors focus on the conditional setting–CVAE and $\sigma$-VAEs. The paper also proposes Calibrated Robust $\sigma$-CVAEs, an approach where $\sigma$ is calibrated to mitigate numerical instability and prevent posterior collapse. The proposed algorithm is analyzed on 1 toy dataset, and also demonstrated on several tabular datasets, density estimation datasets and 2 image datasets.

**Audience:**

Yes

**Claims And Evidence:**

Yes

**Requested Changes:**

* Maybe I missed it, but it would be helpful to explicitly mention the role of each variable, e.g. $y$ for the data distribution, $x$ for the conditioning, $z$ for the latent and etc…
* Can the authors explain how they derived Eq. 7, and what does it mean in practice?
* Please explain the motivation or the idea behind the proposed algorithm in practice–e.g., how the tolerance $C$ is chosen/tuned, how did you come up with the specific heuristic for the calibration update?
* In section 4, can you please explain what it means to generate a dataset with a true $\sigma$? “We simulated three datasets of size n = 5 000 with 2,500 for each class using $\sigma$ = 0.1.” This will explain how you can actually recover it in the experiment.
* Can the authors provide any insights as to what happens in non-conditional settings when using $\sigma$? [non-critical]
* Fix “Minor” from above.

**Strengths And Weaknesses:**

**Strengths**:

* Informative introduction to $\sigma$ and $\beta$-VAEs.
* Goes beyond the usual linear VAEs assumptions, making the theory applicable for more complex, high-dimensional data.
* Interesting insights regarding the biased gradient w.r.t $\sigma$.
* Analysis on the toy dataset provides interesting insights on the training dynamics when interleaved calibration of $\sigma$.
* Comparison with non-parametric density estimators and Flow-based models.
* Detailed appendix.


**Weaknesses**:
* The claim in page 7 that the linear schedule is “computationally expensive” is unclear, and I don’t understand why the authors claim that annealing the KL divergence is a computational strain (“Obviously, these KL annealing techniques are computationally expensive and the statistical insight of the anneal schedule is obscure in principle.”). Also in the Discussion section it is referred to as “expensive”. Given the additional complexity of the proposed algorithm (see below), I don’t understand this claim. Either back this claim up with examples or relax it.
* The proposed algorithm adds additional complexities to Rybkin et al. and introduces another hyper-parameter, the “calibration tolerance” $C$ - which requires tuning, and the calibration is done by using a heuristic (i.e., why sample $\phi$ and increase the variance by 1 after each update?). Is this heuristic somehow related to the theory? Also, in Algorithm 1 there is a condition statement “if Convergence”, what does it mean? From section 4, “convergence threshold is 0.001 in the average loss change”, is that another hyper-parameter?
* The performance on the image datasets is not entirely convincing (but I appreciate the discussion on that in Section 5).

**Minor**:
* Page 2, right after Eq.3, there is this sentence: “To understand the optimality of the parameters, the double inequalities in Eq. 4 is straightforward. “ – I found it very unclear.
* It seems that the citations format is a bit undefined. For example, just after Eq. 8 on page 4, I believe the authors meant to use \citep but the citation used something else.
* Page 7: “A general pseudo-code can be found below.” - I recommend explicitly referring to the algorithm using \ref{}.
* Section 4 - Experiments - missing citations for optimizer (Adam) and datasets (MNIST, CelebA).
* “Calibration tolerance” (section 3.3) = “posterior tolerance” (Algorithm 1) = “convergence threshold” (section 4) ? Please clarify that, because I think that according to Algorithm 1 “convergence threshold” is something else, like an additional hyper-parameter for performing the calibration.
* Just before section 4.1 - “condition image generation” -> “conditional image generation”.
* Section 4.1, please explicitly mention what $K$ is when it first appears (I somehow linked it to Algorithm 1 where it is the number of maximal iterations).
* Figure 1 - “sigma” -> “$\sigma$”.
* Figure 3 - the legend says “EBLO” instead of “ELBO”.
* M1-3 datasets - please refer the reader to the Appendix when you first mention them, as their details are there.

---

> ### Author Response · Authors · 2024-10-08
> **Thank you for your valuable feedback!**
>
> We appreciate your rapid yet thorough review and the suggestions to improve our paper. We have made changes and added additional experiments accordingly. Below, we address all your concerns and questions in an organized manner.
>
> These following questions are related to the understanding of our algorithm from Weekness #2 and Requested Changes:
>
> >The motivation of algorithm
>
> The motivations of the algorithm are to obtain doubly robustness. Namely, our algorithm is proposed to improve the biased and numerically instable estimation of $\sigma$ caused by suboptimal posterior, which is proved in section 3.1, to mitigate the inevitable posterior collapse without introducing $\beta$, which is shown in section 3.2.
>
> >The idea & how did you come up with the specific heuristic for the calibration update?
>
> The idea behind our algorithm is derived from the idea behind KL annealing, which is described in Page 7 above Eq.16: we can find better local optimal parameter through an $\sigma$/$\beta$-annealed loss landscape in a non-convex optimization problem.
>
> Obviously, it is true even if $\sigma$ becomes a parameter that is updated in a two-step iterative algorithm, since we still can anneal the loss conditioning on $\sigma$. If we use explicit cyclical KL annealing schedule shown in Eq.16, we will end up with an explicit $\sigma$ calibration in Eq.17. Likewise, if we calibrate $\sigma$ using our proposed algorithm with calibration in Eq.18, we essentially perform an implicit KL annealing only when posterior collapse is detected at convergence. For an illustrative comparison on explicit and implicit KL annealing, please refer to the tuned $\beta$ in Figure 2 from Fu et al. (2019). and the calibrated $\sigma$ in Figure 2(a) in our paper.
>
> In addition, our algorithm is intentionally designed to perform KL annealing without explicitly using $\beta$ for reliable probabilistic modeling which sticks to ELBO-derived loss and enables the learning of $\sigma$. See section 1.3 for the detailed reasoning.
>
> > how the tolerance is chosen/tuned in practice
>
> The hyper-parameter C is the permissible deviation from a zero KL divergence. An encoder, that has a KL divergence larger than this threshold, is no longer considered having “posterior collapse” issues and won’t trigger our proposed calibration. Therefore, in principle, it should be chosen depending on your requirements on the encoder's performance. We use common definitions following Bowman et al. (2016); Fu et al. (2019).on posterior collapse, e.g. $KL[q_\phi(z|y,x)||p(z)]$ less than a small value C.
>
> In practice, we initialize it to be small number like 0.025 and increase it making encoders to be deviant from have a zero KL divergence. In our ablation experiment, we show in Figure 5(d) that the learned model using C=0.05 yields the almost the same results as the one using C =0.5, in terms of preventing posterior collapse. Of course, one can increase it asking for better encoders with larger KLD. We have discussed other possible choice of definition of posterior collapse in Discussion section as well as our thoughts in Appendix F.
>
> >Is this heuristic somehow related to the theory
>
> Our algorithm is just one heuristic realization of solutions to tackle the limitations of VAE approximation shown in Theorem 2.4, i.e. the non-identifiability of $\sigma$ and the non-guaranteed optimal encoder. The soundness of our algorithm is rooted in the empirical success of KL annealing method. In particular, our analysis in section 3.2 reveals the fact that the existing successful KL annealing methods can be seen as decoder variance calibration via $\sigma$-$\beta$ equivalence. If you look for a more rigorous theoretical result for our algorithm, you might need additional assumptions. See the first paragraph of section 3.2 for our thoughts on constructing a rigorous theory on breakdown points.
>
> >Why sample $\varphi$ and increase the variance by 1 after each update
>
> We sample $\varphi$ as calibration can upscale or downscale. The variance of $\varphi$ determines the magnitude of scaling to the $\sigma$. To see it, the half-normal distributed multiplication factor $|1+\varphi|$ in Eq. 18 falls into a limited range [0,2.96] if $S=1$ with 95% probability, which expands as $S$ increases.
>
> An increased magnitude of the relative changes of $\sigma$ during the calibration process expands the exploring range of the calibrated $\sigma^{cal}$. This is extremely helpful if the estimated $\sigma$ is frequently trapped in certain stationary points as shown in Figure 2(a) or if you intialize $\sigma$ at a small value, following the suggestion from Lucas et al. 2019a, and want to effeciently calibrate $\sigma$ to be a large value that possibly related to a smoother loss landscape.

---

> ### Author Response · Authors · 2024-10-08
> **Thank you for your valuable feedback! (Continued)**
>
> >**“Why annealing the KL divergence is a computational strain” (Weakness #1)**
>
> The explicit KL annealing is undoubtedly easy and simple to implement. However, it is less efficient and often requires more computational power in terms of preventing posterior collapse.
>
> For example, if $\sigma$ is assumed to be fixed, an explicit KL annealing schedule like Eq.17 is essentially a series of deterministic $\sigma$ calibrations in Eq.16, due to $\sigma$-$\beta$ equivalence. In other words, the annealing process of loss landscape is predefined and not adaptive to the performance of encoders. Similarly, when $\sigma$ is a learned parameter, an explicit KL annealing schedule confounds with it in shaping the loss landscape, similarly to how it would by a single “effective” $\sigma_e$. For example, when $\beta=0.25$ adding to LHS of Eq.(5), the “effective” $\sigma_e$ is half of current $\sigma$ value, i.e. $\sigma_e = 0.5*\sigma$ . The confounding between $\beta$ and $\sigma$ will not vanish unless $\beta$ is 1. In extreme cases, if the predefined $\beta_t =  1/(2\sigma_t^2)$, the effective $\sigma_e$ would be fixed at 1. This means that there would be no KL annealing to the loss landscape at all because there would be no change in the effective $\sigma_e$.
>
> Such confounding not only complicates the annealing process of loss landscape determined by the effective $\sigma_e$, but also hinders the accurate and efficient estimation of $\sigma$ under maximum likelihood principle. Therefore, we argued in the end of section 3.2 that it should be done more efficiently by more direct calibrations on a learned $\sigma$. Our last contribution shown in Introduction is to confirm that Occam's razor applies to it – preventing posterior collapse via efficient $\sigma$ calibrations can be as few as we need without introducing $\beta$.
>
> To empirically verify the above theory, we trained Gaussian $\sigma$-CVAE with a learned $\sigma$ using a monotonic or a cyclical annealing schedule tuning $\beta$ linearly from 0 to 1. For cyclical annealing, we use a cycle period $M=10$ of 5 maximum cycles . For monotonic anneal, we consider a single monotonic annealing with same length $M*5=50$. The $\beta$ becomes a fixed value of 1 after the maximum 50 iteration reached untill the algorithm converges. The hyper-parameters of KL annealing are chosen reasonably based on what we observe in Figure 2(a) for a fair comparison.
>
> We report the MSE of learned $\sigma$, KLD at convergence under the same setting of Figure 3 average of 20 repetitions. Training Loss is not reported because of the training loss inconsistency that is caused by using different $\beta$.
>
> | $\sigma$ Initialization | 0.2  |  | 0.4 |  | 0.6 |  | 0.8 |  | 1 |  |
> |---|---|---|---|---|---|---|---|---|---|---|
> |  | MSE | KLD | MSE | KLD | MSE | KLD | MSE | KLD | MSE | KLD |
> | Monotonic | 1.62e-02 | 1.81 | 6.09e-03 | 2.10 | 5.65e-03 | 2.19 | 3.41e-04 | 2.12 | 6.06e-04 | 2.17 |
> | Cyclical | 1.76e-02 | 1.77 | 3.24e-02 | 1.71 | 1.92e-02 | 1.84 | 2.17e-02 | 2.01 | 4.58e-02 | 1.45 |
> | Calibrated C=0.05 | 2.41e-02 | 1.79 | 2.71e-02 | 1.64 | 6.01e-03 | 1.92 | 5.38e-02 | 1.39 | 2.69e-02 | 1.70 |
> | Calibrated C=1.96 | 1.26e-05  | 2.09 | 5.77e-05 | 2.09 | 1.22e-05 | 2.08 | 1.73e-05 | 2.10 | 1.76e-05 | 2.09 |
> | Vanilla | 0.16 | 4.17e-01 | 0.20 | 1.40e-03 | 0.17 | 3.16e-01 | 0.20 | 2.20e-04 | 0.20 | 5.39e-04 |
>
> We also report the maximum wall time till convergence under the same setting of Table 4 using $K=25,50,100$ recorded by python function time.perf_counter() out of 5 successfuly repetitions.
> |  | K=25 | K=50 | K=100 |
> |---|---|---|---|
> | Monotonic | 24.85  | 36.26  | 58.38 |
> | Cyclical | 10.76 | 23.01 | 45.59 |
> | Calibrated C=0.05 | 10.55 | 12.69  | 25.00 |
> | Vanilla | NA  | NA | 62.15 |

---

> ### Author Response · Authors · 2024-10-08
> **Thank you for your valuable feedback! (Continued)**
>
> >The proposed algorithm adds additional complexities to Rybkin et al. and introduces another hyper-parameter, C (Weakness #2)
>
> Rybkin et al. itself is an extremely simple two-step iterative algorithm for estimating $\sigma$-VAE, where the decoder variance $\sigma$ is updated under maximum likelihood (ML) principle while the update of $\theta,\phi$ in decoder/encoder networks remains to be gradient descent.
> Shown in Eq.10 of Rybkin et al, the ML estimation of decoder variance is efficiently fast via an analytical closed form, particularly in cases when the decoder variance is assumed to be a scalar scaled identity matrix.
>
> We added these complexities step by step in our paper, with clear motivations to tackle the limitations of obtaining tighter approximation gap and detailed statistical interpretations. Since our algorithm includes two meaningful complexities, namely a semi-coordinate descent of $\theta,\phi$ with inner step $K$ and the calibration on $\sigma$ with hyper-parameters $C$, we must admit that they increase the computation cost respectively.
>
> However, compared to explicit KL annealing, our calibration itself is directly calibrates $\sigma$ when observing posterior collapse and the estimation of $\sigma$ remains to be as efficient as Rybkin et al. As we stated in the rebuttal above, it neither requires an additional hyperparameter $\beta$ nor sophisticates the estimation process of $\sigma$.
>
> More importantly, we highlight that the combination of these two mechanisms is beneficial to reduce computational cost. In other words, our algorithm is robust in small $K$ settings, reducing the need for a full coordinate descent on $\theta,\phi$ or the need for an excessively large $K$ to provides a better estimation in Figure 1(a). For instance, we showed the calibration significantly reduces the max time to convergence for $K$ ranging from 25 to 500 in Table 4 in Appendix E, while constantly providing a robust estimate of $\sigma$.
>
> Let us know if the additional explanations and experiments are enough to enhance your understanding of (cyclical) KL annealing's performance improvements via the (periodic) tuning of $\beta$ from 0 to 1, and to improve your review on our double robust algorithm via $\beta$-$\sigma$ loss equivalence.
>
> >“The performance on the image datasets is not entirely convincing (but I appreciate the discussion on that in Section 5).” (Weakness #3)
>
> This is indeed one of the major limitations of our algorithm. We have suspected several reasons as we stated the limitations of our algorithm. In addition, at the data level, CelebA dataset have not only label corruption but also extremely diverse background pictures within popular face classes(even after center chopping), which might also hinder the human face generation and the robust estimation of $\sigma$.
>
> > Miscellaneous questions: Can the authors explain how they derived Eq. 7, and what does it mean in practice?
>
> Eq.7 is a direct result of location-scale factorization on the decoder, following the model assumption on the mutual independence between covariates $X$, latent variable $Z_1$, and decoder variance $Z_2$. Notice that the Gaussian decoder distribution, given latent variable $Z_1$ is a location scale distribution. In other words, $Y|(X=x,Z_1=z_1) = \mu_\theta(x,z_1)+\sigma*Z_2$. Due to the above independence, we can get $(X,Y) = (X, G_{\theta,\sigma}(X,Z_1,Z_2))$ in distribution leading to Eq.7.
>
> In practice, Eq.7 states the structure of randomness of the generative model and its sampling process. Gven any covariates $X$, you can sample from the learned distribution of response variable $Y$ via reparameterization trick, using independently latent sample $z_1$ and decoder noise sample $z_2$. Eq.7 also demonstrates that the VAE generative model is an infinite Gaussian mixture as the latent variable, serving as the prior of the decoder mean, may not be finitely valued.

---

> ### Author Response · Authors · 2024-10-08
> **Thank you for your valuable feedback! (Epilogue)**
>
> > In section 4, can you please explain what it means to generate a dataset with a true $\sigma$? “We simulated three datasets of size n = 5 000 with 2,500 for each class using $\sigma$= 0.1.” This will explain how you can actually recover it in the experiment.
>
> Sorry for the notation abuse. The $\sigma=0.1$ refer to the variance of $\epsilon_i$ in Section 4.1 for generating two-moon dataset. We are going to call it $\sigma_0$. This two-moon dataset is a oracle non-linear VAE model with $\sigma_0=0.1$ and  $\mu_0(x,z_1)=(cos(\pi\Phi(z_1)-\frac{1}{2}x, (-sin(\pi\Phi(z_1)+\frac{1}{6})x)$, where $\Phi$ is the c.d.f of a standard normal. We sample $\alpha$ from a Uniform distribution instead of computing $\pi\Phi(z_1)$ for convenience.
>
> The $\sigma$-Gaussian CVAE model should be expressive enough as shown in Theorem 2.4, to patten match both the variance of $\epsilon_i$ with $\sigma=0.1$, and the non-linear mean function using neural networks. Meanwhile, it is challenged by posterior collapse and biased estimation of $\sigma$. Refer to Figure 4(a) in Appendix E to see what the dataset looks when $\sigma_0=0.1, 0.2,$ or $0.3$.
>
> >“Calibration tolerance” (section 3.3) = “posterior tolerance” (Algorithm 1) = “convergence threshold” (section 4)?  Clarify the condition statement “if Convergence”, what does it mean?  From section 4, “convergence threshold is 0.001 in the average loss change”, is that another hyper-parameter?
>
> Sorry for the inconsistent notation. Yes. the convergence threshold is the hyper-parameter that is needed for the algorithm to stop the training process, which exists ubiquitously in iterative optimization methods, and is different to calibration/posterior tolerance. The latter two both refers to the hyper-parameter $C$, which are abused in concept since calibration is based on a posterior collapse metric. We have made the notation consistent.
>
> The convergence is the first-step criterion to stop our proposed algorithm without an early stopping mechanism. When the average negative Elbo on training dataset does not changes larger than the convergence threshold, which is a pre-defined value. We will perform one of following actions. We will break the training process iterations if we do not detect the posterior collapse of decoder. Otherwise, we trigger our calibration process on $\sigma$.
>
> **We have made the changes regarding the minor issues you mentioned. In addition, we improved the readability as shown below:**
>
> 1.Improving citation format in several locations.
>
> 2.Page 2, right after Eq.3, there is this sentence: “To understand the optimality of the parameters, the double inequalities in Eq. 4 is straightforward. “ $\to$ “The double inequalities in Eq. 4 provides a straightforward insight into model parameters’ optimality. “
>
> 3.Page 7, before sec. 3.3, “Taking Eq.17 as an example, ….” $\to$ “Taking Eq.16 as an example, ….” (Eq.17 itself is the result)
>
> 4.Page 8, after sec.4.1 “In other words, a well behaved encoder is supposed to have a zero KL divergence” $\to$ “In other words, a well-behaved encoder should not exhibit a zero KL divergence”  (missing not added)

---

> > ### Comment · Reviewer_gj1z · 2024-11-30
> > **Thank you for your effort**
> >
> > I appreciate the authors response, corrections and additional experiments, thank you for the effort you made during this period. Most of my concerns have been addressed (I would still apprciate an answer to the question: "Can the authors provide any insights as to what happens in non-conditional settings when using $\sigma$?"). There are still typos in Figure 5 ("EBLO" -> "ELBO").

---

### Review · Reviewer_kWBh · 2024-10-27

**Summary Of Contributions:**

SUMMARY OF PAPER

1.1 The authors study the twin problems of learning and inference in $\sigma$-conditional variational autoencoders ($\sigma$-CVAEs) whose latent variable has a data-independent prior.

1.2 The loss function for these architectures is the negative ELBO. The authors highlight two issues when optimizing this loss function. The first is the quality of the variational approximation used for inference; the second is the expressiveness of the generative model used for learning. The authors argue that these issues must be examined in tandem to be fully understood.

1.3 The authors review the relationship, previously reported, between a $\beta$-CVAE (where $\beta$ is an annealing parameter) and the $\sigma$-CVAE with a fixed unit variance $\sigma^2=1$.

2.1 The authors review an earlier result (Agrawal & Domke) that the randomness of any distribution can be "outsourced" to a Gaussian random variable, and use it to show that a $\sigma$-CVAE can in principle express an arbitrarily complex distribution in the limit $\sigma\rightarrow 0$.

2.2 The authors review a previous result (Sobol) on block neural decomposition, then show how with related assumptions $\sigma$-CVAEs can provide the same guarantee in (what they call) the *non-asymptotic* regime when $\sigma$ is finite but nonzero (Theorem 2.4).

3.1 The authors review that the gradient of the ELBO is a biased approximation to the gradient of the model's true log-likelihood, and that this bias extends to gradient-based (and other types of) estimates of $\sigma$.

3.2 The authors argue that posterior collapse is a sign of a poor encoder, and that it should be used to anneal/calibrate the variance $\sigma$ in $\sigma$-CVAEs (and not merely the hyperparameter $\beta$ in $\beta$-CVAEs.) They also show how a previous schedule for $\beta$-annealing can be converted into a schedule for $\sigma$-annealing.

3.3. The authors argue that the annealing schedule for $\sigma$ should be tied to evidence of posterior collapse (rather than fixed and pre-determined in advance), and they propose a new calibrated schedule that does so.

4.1 The authors show that $\sigma$-CVAEs with this calibrated annealing schedule can learn the two-moon data set with a proper batch size K and calibration hyperparameter C. They argue that their method is *doubly robust* because (1) it works with varying initializations of $\sigma$, and (2) it is better at avoiding posterior collapse.

4.2 The authors report that their $\sigma$-CVAEs outperform four other methods (WGCS, NNKCDE, CKDE, basis expansion) on three additional data sets.

4.3 The authors compare calibrated $\sigma$-CVAEs to Bayesian normalizing flows on 6 UCI data sets. The comparisons are based on log-likelihoods, which disadvantages the $\sigma$-CVAEs, since they only compute lower bounds on likelihoods. With this in mind, the results for calibrated $\sigma$-CVAEs appear to be competitive.


SUMMARY OF CONTRIBUTIONS

The main contributions, as I see them, are the specific proposal (section 3.3) to anneal the variance parameter in $\sigma$-CVAEs based on evidence of posterior collapse, rather than with a pre-determined annealing schedule, and the experimental results (section 4) showing that this strategy is effective. Theorem 2.4 also appears to be novel, but its meaning and significance were less clear.

**Audience:**

Yes

**Broader Impact Concerns:**

None.

**Claims And Evidence:**

No

**Requested Changes:**

The writing needs improvement throughout. The authors should also address the following points.

* The authors stress in sections 1 and 2 that a zero approximation gap is achievable without perfect variational inference. From the writing, it seems that the authors find this to be surprising. But it is unclear why this should be surprising. A zero approximation gap is possible when the generative model is sufficiently powerful to parameterize (express) a distribution that perfectly fits the data. The expressibility of the generative model is not related in any way to the difficulty of approximating its posterior by variational inference. (Indeed, that is why the proofs of section 2 do not make any reference to the variational approximation, $q$, for the encoder.)

* There are many possible connotations that can be attached to the words *robust* and *asymptotic*. For this reason, the authors should be as explicit as possible about their meaning when they first introduce them.

* The authors should not use $q$ to denote both the variational approximation in section 1.1 and the variable dimensionality in Defintion 2.2.

* I hope that the authors can expand (or clarify) the discussion of identifiability in section 2.2. I found the assumptions behind Theorem 2.4 to be somewhat unintelligible. Also, is there any connection here to the relationship between posterior collapse and latent non-identifiability discussed in the NeurIPS-2021 paper by Wang, Blei, and Cunningham?

* The authors stress in 3.1 that gradient-based (or coordinate-descent-based) updates of the ELBO are biased because they are designed to maximize the ELBO rather than the true likelihood. I believe this bias is well-known and widely appreciated in all models where the ELBO is maximized. It was not clear what special relevance this has for $\sigma$-CVAEs, or why the authors devoted so much space to this point.

* More generally, I often did not see the connections that the authors were attempting to make between different sections of the paper (e.g., sections 2 and 3, sections 3.1 and 3.2).

* The authors provide pseudocode in Algorithm 1, but this pseudocode does not include the specific form of the calibration as given in eq. 18. It seems to be a strange omission.

**Strengths And Weaknesses:**

The main strength of this paper is its algorithmic contribution: it proposes a specific annealing schedule for the variance parameter in $\sigma$-CVAEs, and it demonstrates experimentally that this annealing schedule---which is not fixed in advance but triggered by evidence of posterior collapse---leads to better learning. There is also a theoretical contribution--a proof that under certain assumptions, $\sigma$-CVAEs are expressive enough with a nonzero value of $\sigma$ to parameterize an arbitrarily complex distribution. Discussions throughout the paper attempt to provide a coherent picture of inference and learning in $\sigma$-CVAEs.

The main weakness of the paper is that it is poorly written at many levels. The following are some areas of the writing that need improvement:

* The introduction does not begin by providing a high-level overview. I encourage the authors to give one or more talks on this work; I think it will force them to present the big picture first. It's unusual for a paper to have five equations by the bottom of page two. It should be possible to discuss the bigger issues of non-identifiability, posterior collapse, and inference/approximation gaps in VAEs without first introducing (nearly) all of the paper's notation.

* Individual sentences are poorly balanced. For some examples, see the first sentence of the abstract, or the first sentence of 3.2.

* The logical connectives between sentences do not always flow: e.g., the phrase "in other words" should only be used to introduce an idea that is an obvious recapitulation of what comes before.

* The authors do not seem to understand the difference between in-text (\citet) and parenthetical citations (\citep). Some passages of text are nearly unreadable.

The above list is not exhaustive, and the cumulative effect of these issues is to place an undue burden on the reader. A paper should not require several passes to understand its main points.

---

> ### Author Response · Authors · 2024-11-01
> **Thank you for the review!**
>
> Thank you for the review! We appreciate your comprehensive review and suggestions to improve our paper, and we are encouraged by your recognition of our proposed algorithm as a notable strength.
>
> In summary, we have revised our manuscript to address your primary concerns regarding both the clarity of the writing and the presentation of our theoretical contributions.
>
> * Below, we provide itemized feedback to your concerns regarding the writing:
>
>  >The introduction does begin by providing a high-level overview
>
> In the introduction, we do provide a short overview following the introduction of each notation in sec 1.2 and 1.3. Per your suggestions, We have added an extended Related Work section in the appendix and referenced it in the main text. To provide a big picture, the Related Work section is consists of 3 talks, including expressive decoders and approximation gaps, model non-identifiability and robust estimation of decoder parameter, posterior collapse and KL annealing tecnniques.
>
> >Individual sentences are poorly balanced; The logical connectives between sentences do not always flow
>
> We have revised these sentences for clarity, including the first sentence of the abstract. The logical flow has been refined in multiple sections. We hope the revision enhances readability; please let us know if further adjustments would be beneficial.
>
> >The authors do not seem to understand the difference between in-text (\citet) and parenthetical citations (\citep).
>
> Thank you for pointing it out. It is also mentioned by another reviewer, and we have corrected it throughout. Please refer to the revised version for these updates.
>
>
> >There are many possible connotations that can be attached to the words robust and asymptotic. be as explicit as possible about their meaning<
>
> We have highlighted the asymptotic as taking $\sigma\to0$ and provided an explicit definition of robustness within our context.
>
> >The authors should not use to denote both the variational approximation in section 1.1 and the variable dimensionality in Defintion 2.2<
>
> We believe there is no confusion in our notation, as the variational approximation is consistently denoted as q_\phi and the response variable dimension as q.
>
>
> * The rest of the rebuttal aims to improve your understanding of the meaning and significance of our theoretical contributions.
>
> >Theorem 2.4 also appears to be novel, but its meaning and significance were less clear. Why the authors find this to be surprising that “A zero approximation gap is achievable without perfect variational inference"?
>
> Our results are indeed not surprising and align with existing theories and intuition, but we emphasize that a rigorous statement of such claim is not trival and the word “generally” should not be omitted. Specifically, we highlight our contributions in the following aspects.
>
> Contextualization: Our theories reveal the fundamental assumptions of Gaussian $\sigma$-CVAE as a latent variable model compared to a Gaussian noise outsourced generating map. It provides another angle in understanding the existing learning theories. For example, in Lemma 2.1, we show how the asymptotic assumption in Dai and Wipf 2018 that taking $\sigma\to 0$ can be understood in Eq.7. i.e., the data generating distribution is approximated by the mean function $\mu_\theta(x,z)$ only.
>
> Approach and methodology: Our theories leverage functional ANOVA analysis and focus on Gaussian $\sigma$-CVAEs, which is a relatively less explored topic. In the paragraph following Theorem 2.4 in page 5, we argue that our characterization eliminates two key assumptions in existing theories regarding the decoder, which we find impractical: (1) assuming $\sigma\to0$, and (2) the assumption in linearity VAEs. In our opinion, these assumptions are unrealistic simplifications for achieving near-optimal variational inference.
>
> Prevalent significance: Our theoretical results better align with practical scenarios including our experimental results in Section 4.1, where the model recovers the data-generating distribution that is a nonlinear transformation of latent without taking $\sigma \to 0$ or having an optimal variational inference. Such experiment results can’t be fully explained in linear VAE theories starting from Lucas et al NIPS 2019b or in the line of asymptotic analysis in Dai and Wipf 2018.
>
> For a rigorous discussion, we are open to discussing and citing any related literature on the approximation gap of $\sigma$-VAEs/CVAEs that we may have overlooked. Please let us know if there is any related literature that you think is relevant.

---

> ### Author Response · Authors · 2024-11-01
> **Thank you for the review!**
>
> >The expressibility of the generative model is not related in any way to the difficulty of approximating its posterior by variational inference (Indeed, that is why the proofs of section 2 do not make any reference to the variational approximation, q, for the encoder.)
>
> We respectfully disagree with this comment.
>
> In our opinion, a limited decoder can simplify the true posterior, making it easier for the encoder to approximate. In mathematical terms, inference gaps for any $\phi$ is a function of $\theta,\sigma$. As we stated in the introduction 1.2, unnecessary assumption that limits the expressibility of generative model is wildly used for VAE theories. For example, linear VAEs assume that decoder‘s mean is a linear combination of latent variable, resulting in a conjugate gaussian posterior that is easy for gaussian encoders to approximate. See (Lucas et al NIPS 2019b) for a closed form expression of true posterior.
>
> Our proof does not rely on $q_\phi$ because our theoretical framework focuses solely on the expressiveness of decoders, which allows for suboptimal or poor encoders. The correct interpretation of our theorem is that “although a more expressive of the generative model is often leads to a higher difficulty of having optimal variational inference, the recovery of data-generating distribution does not necessarily require an optimal encoder as shown in Theorem 2.4”.
>
> Nonetheless, Section 2 is a coherent yet rigorous story of “a zero approximation gap is achievable without perfect variational inference” for $\sigma$-CVAE wherein $\sigma$ is a learned parameter. Notably, practical variational inference improving techniques, such as KL annealing and importance sampling [1][2][3], lack rigorous discussions on their assumptions and expressiveness of the generative model.
>
> [1] Burda, Yuri, Roger Grosse, and Ruslan Salakhutdinov. "Importance weighted autoencoders." arXiv preprint arXiv:1509.00519 (2015).
>
> [2] Nowozin, Sebastian. "Debiasing evidence approximations: On importance-weighted autoencoders and jackknife variational inference." International conference on learning representations. 2018.
>
> [3] Zimmermann, Heiko, et al. "Nested variational inference." Advances in Neural Information Processing Systems 34 (2021): 20423-20435.
>
> >Expand (or clarify) the discussion of identifiability in section 2.2
>
> There are two identifiability issues in our paper.
>
> The first one is the non-identifiability/non-uniqueness of block neural decomposition for any data generating transformation $G^*$, which is solved by the block ANOVA representation in section 2.2, resulting in a unique decomposition that is a sum of orthogonal functional bases. Such decomposition is necessary and is closely related to $\sigma$-CVAE model in non-asymptotic regime where $\sigma$ can't goes to 0. We established an equality condition via patten matching, demonstrating a zero approximation gap.
>
> The second one is the identifiability of model parameters, which is a fundamental challenge in generative models. The definition is that, given unlabelled data $x$, there exist distinct parameters $ \theta_1 \neq \theta_2$ such that $p_{\theta_1,\sigma_1}(y|x) =p_{\theta_2,\sigma_2}(y|x)$. See the definition Eq.4 in [1] and check Ding Zhou, Xue-Xin Wei NIPS 2020 for a solution.
>
> [1] Khemakhem, I., Kingma, D., Monti, R., & Hyvarinen, A. (2020, June). Variational autoencoders and nonlinear ica: A unifying framework. In International conference on artificial intelligence and statistics (pp. 2207-2217). PMLR.
>
> Such issue of non-identifiability of $\theta, \sigma$ is not solved in our paper. Below we rigorously show why it exists in Gaussian $\sigma$-CVAE model $ y =\mu_\theta(x,z_1)+\sigma z_2$. Given any $\theta_1$, consider any vector rotations T(), we have $\mu_{\theta_1}(x,z_1)= \mu_{\theta_1}(x,T(z_1))$ in distribution. The parameter $\theta_2$ is then defined by introducing an additional transformation layer $G()$, which applies an identical transformation to $x$ and a rotation T() on $z$. Consequently, we have $ p_{\theta_1,\sigma}(y|x) = p_{\theta_2,\sigma}(y|x)$ for having same mean parameter $\mu$ and variance $\sigma^2$.
>
> In our paper, we follow the convention assuming $\sigma$ to be a latent independent parameter to reduce the model non-identifiability. If $\sigma_\theta(z)$ is assumed, then $\theta$ that parameterizes $\sigma()$ also becomes non-identifiable using the same argument above.
>
> Nonetheless, in a non-identifiable model, multiple parameter values can produce the same likelihood, resulting in multiple (local) optimal maximum likelihood estimates. In section 3, we leverage this characteristic to anneal the training process using a likelihood-based objective, allowing us to reselect the local optima $\theta, \sigma, \phi$ through our proposed calibration method based on the performance of $\phi$.

---

> ### Author Response · Authors · 2024-11-01
> **Thank you for the review**
>
> >Difference between model non-identifiability to latent non-identifiability by Wang et. al
>
> We have discussed what model non-identifiability is above. Here we discuss the connections to the referrence you mentioned.
>
> On the other hand, the Definition 2 in [Wang et. Al NIPS 2021] is a less used notation that is not related to the model identifiability of marginal likelihood. In their work, they considered if decoder $p(y|x,z)$ in our context (or $p(x|z)$ in a unsupervised task) to be homogenous w.r.t. $z$, and referred to it as the “likelihood”.
>
> However, in our view, this is not a likelihood function in any sense, because you can’t observe z in an unlabeled or labeled dataset. Their notation of non-identifiability leads to what we call “true posterior collapse”, which essentially presents a rare degradation of latent variable model when the decoder becomes independent of latent information.
>
> Therefore, “True posterior collapse” exemplifies an extreme scenario where “a limited decoder can simplify the true posterior for an encoder to approximate” as we mentioned in the rebuttal earlier. Notably, “True posterior collapse” accounts for, if any, small part of approximate posterior (encoder) collapse. The latter is what we refer to as 'posterior collapse' in this paper, a term that is more widely recognized in Lucas et al NIPS 2019 and Razavi et al ICLR 2019.
>
> >Intuition behind the assumptions behind Theorem 2.4 to be somewhat unintelligible
>
> We have revised this section for clarity. Assumption 2.3 is the assumption for the uniqueness of functional decomposition.
>
> The assumptions in Theorem 2.4 are sufficient conditions for achieving pattern matching between ground truth map $G^*$ and $\sigma$-CVAE model. For a non-zero $\sigma$, the only restriction on Gaussian $\sigma$-CVAE is the generalized linear form specified in Eq.7 and the independence between latent $z_1$ and noise $z_2$, allowing for the reparameterization of decoder mean and decoder variance. Thus, similar assumptions should be added to the unique decomposition of data generating map shown in Eq.8, enabling patten matching. Such assumptions reflect the limitation of $\sigma$-CVAE to approximate a data generating distribution that may be arbitrarily complex.
>
> >Biased gradient: what special relevance this has for $\sigma$-CVAE
>
> It highlights another problem that this paper aims to solve: biased parameter estimation using ELBO as loss function instead of the exact likelihood.
>
> For $\sigma$-CVAE, we emphasized that if $\sigma$ is not fixed, the bias in $\sigma$ estimates is related to suboptimal variational inference. In section 3.1, we showed the bias of gradient of $\sigma$ and argued that optimizing $\sigma$ in [Rybkin et al. ICML 2021] is just coordinated descent with no calibration included (although they called it as calibrated). It completes our motivation for proposing new algorithm of $\sigma$-CVAE: 1) improving variational inference 2) robust estimation of decoder parameters, particularly $\sigma$.
>
> >The authors provide pseudocode in Algorithm 1, but this pseudocode does not include the specific form of the calibration as given in eq. 18. It seems to be a strange omission.
>
> We have added the reference of Eq.18 in the algorithm. The pseudocode in Algorithm 1 is, as the title denoted, a general $\sigma$ calibration-at-convergence framework extending [Rybkin et al. ICML 2021]. We believe the details of calibration are less important than the idea that an effective calibration should be less ad-hoc and more adaptive to the performance of encoders. Thus we intentionally keep it at a high level to encourage reasonable modifications from experienced readers. In the Discussion section, we have also discussed how to select alternative criteria for posterior collapse, acknowledging that the definition of posterior collapse may vary.

---

> ### Comment · Reviewer_kWBh · 2024-11-29
> **Reply to author replies**
>
> Thank you for the detailed reply and many clarifications.
>
> There are two different ways to discuss the expressiveness of the model. One way is to ask: what distributions can the model parameterize if (say) its parameters are set by an oracle? Another way is to ask: what distributions is the model able to learn if the learning procedure involves an approximate step of variational inference?
>
> An answer to the first question does not require any reference to VI (and it seemed to me that your proof was of this nature).
>
> The second question is both more interesting and more difficult to answer. From your reply, it seems  that you are interested in this second question. I agree that the "effective" expressiveness of the model, as it is estimated in practice using VI, may depend on the nature of the VI approximation. I'm not entirely clear how your results address this question.
>
> I would also be interested in reading your replies to the third (and last-submitted) review.

---

> ### Comment · Reviewer_kWBh · 2024-11-30
> **Revised manuscript**
>
> The writing is much improved in the revised manuscript.
>
> One typo that I found: p.3, 2nd paragraph, "parametersunbounded"

---

### Review · Reviewer_sQRH · 2024-11-15

**Summary Of Contributions:**

The paper investigates VAEs in which the decoder variance is a scaled identity matrix $\sigma^2 I$. The object of the paper is to find ways to train the VAE, with particular attention given to $\sigma$. The authors want (i) accurate estimates of $\sigma$ and (ii) no posterior collapse (that is, the variational approximation $q$ reduces to the prior distribution $p(z)$). The authors propose an algorithm, which alternates optimizing the variational parameter $\phi$, the decoder's weights $\theta$, and the decoder's variance $\sigma$. The algorithms departs from standard optimization by introducing a "calibration" step, which increases $\sigma$, if there are signs of posterior collapse. The method is compared to annealing methods, which rely on an inverse-temperature $\beta$. The authors argue that their method is easier to tune and apply it to several examples.

**Audience:**

Yes

**Broader Impact Concerns:**

No concerns.

**Claims And Evidence:**

Yes

**Requested Changes:**

See the recommendations above.

I have some minor comments:
- In Table 1, what does the crossed checkmark mean?
- p7, the authors write "it collapse to a nonlinear regression". Do you mean "linear"?
- p7, define mod.
- p7 don't use $C$ for tuning parameter, since it is used for condition (2) of theorem 2.4
- p10 can you show the optimization paths for multiple initializations?
- p11 provide a summary of the models in the main body, even if it's only one sentence per model, and indicate the dimension of each model. The reader should understand the scope of the experiments, without reading the appendix.
- p12 can you recommend starting defaults for $K$ and $C$?

**Strengths And Weaknesses:**

The proposed algorithm is intuitive and simple to implement. It is not entirely clear how to choose the calibration tolerance $C$ and the number of steps $K$, although the experiments provide insight for the latter tuning parameter. I would agree that the method is easier to tune than annealed VAE, although I wonder if it is less flexible. The authors demonstrate the benefits of the calibration step, showcasing its ability to recover the true $\sigma$ in a simulation setting.  I would suggest moving the comparison with KL annealing to the main body and highlighting the results in Table 5.

The theoretical motivation is nice. I enjoyed the discussion about the inference and the approximation gap (although I believe it can be improved), and the equivalence between $\sigma$-CVAE and $\beta-CVAE. I believe Sections 1.2 and 1.3 can be more concise.

Equation (4) plays a central role but, as is, it is difficult to decipher. I would indicate that the inference gap is given by the KL divergence between $q_\phi$ and $p(z \mid x)$. I'd also have a preference for flipping the inequalities, and working with the ELBO rather than the negative ELBO, just to agree with the fact that the ELBO is a lower-bound and that typically we try to maximize the evidence.

Section 2 seems interesting but lacks clarity. I could not understand the main point, namely that $\sigma$-CVAE approximates arbitrary densities, even when the variational inference is suboptimal and the decoder non-identifiable. My questions are the following:
* how restrictive is condition (2) in Theorem 2.4 (note the typo in the first term of the neural block decomposition: it should be $f^\eta_{z_1 z_2}(z_1, z_2)$ instead of $f^\eta_{x z_2}(z_1, z_2)$)?
* can the authors indicate which terms depend on $q_\phi$? Is there an optimal solution for any variational parameter $\phi$?

I'm not convinced by the title of Section 3.1. "The necessity of $\sigma$-calibration: bias in the gradient". The main argument seems to be that our estimate of $\sigma$ is biased, because we minimize the ELBO instead of the evidence. While this indeed leads to a bias estimator, I don't understand how this motivates $\sigma$-calibration. On the other hand, I find the discussion in 3.2 and the connection to annealing more compelling motivation for $\sigma$-calibration.

In 3.3, I disagree with the authors' claim that their objective sticks to the original ELBO loss and avoids explicit use of $\beta$. First, as the authors note, there is an equivalence in loss and secondly, algorithm 1 does not quite define an optimization algorithm, since the calibration step is a departure from optimization. Framing it as a minimizer of the ELBO is misleading, and annealing has the merit of making the change in loss function explicit. By contrast, it is not clear which objective function, if any, the proposed algorithm minimizes.

In 4, the experiments are interesting and demonstrate the benefit of adding a calibration step, as well as comparing different calibration strategies. Still, I have several reservations:
* How much sense does it make to compare $\sigma$ between different models? In Figure 1, right (a) and (b), the performance of vanilla $\sigma$ -CVAE would be worse if we reduced $\sigma$ to 0.1. The large $\sigma$ accounts for variability which is unexplained by the model. Moreover, $\sigma$, on its own is not a meaningful quantity, it only exists in the context of a model. What would be more interesting is to compare uncertainty calibration, for example using a score function on held-out data.
* Similarly, I'm not sure I understand the comparison of SD in Table 2. Why is a lower SD better?
* Could the figures, including Figure 1, account for the varying cost per iteration, as $K$ varies?
* I don't understand the takeaway of Table 2. It seems $\sigma$-CVAE underperforms. The authors write "our method is very competitive with other methods" but this qualitative statement is vague. What would an uncompetitive algorithm look like?

Appendix A should appear in the main body.

The writing of the paper can be improved:
* there are numerous syntax errors and typos. Please proofread the manuscript.
* there are too many vacuous words, such as "complex" or "reliable". How do the authors define "reliable"? Similarly, the authors make extensive use of the word "robust", but usually do not specify "robust" to what. It is often unclear whether the authors are talking about posterior collapse, or poor estimates of $\sigma$ (although my point above about how meaningful $\sigma$ is on its own...).

---

### Decision · Action_Editor_ZZwW · 2025-01-05

**Recommendation:** Accept with minor revision

**Comment:**

Reviewers overall liked the idea of the paper being simple and relatively easy to implement. They appreciated the mathematical analysis regarding sub-optimality of variational inference and how that relates to posterior collapse. During discussion period the paper was improved with better presentation and additional motivating examples.

I recommend the following revision actions for camera ready:

1. As per author-reviewer sQRH discussions: a further clarification of the approximation theory. (https://openreview.net/forum?id=VIkycTWDWo&noteId=NPhSjt6Cq8)

2. Clarify the discussion on expressiveness, and how is this related to posterior collapse.

**Audience:**

Probabilistic ML researchers, especially for those interested in variational auto-encoders (VAEs).

**Claims And Evidence:**

This paper proposes new training methods for VAE, with particular attention on (1) accurate estimation of the decoder output variance, (2) avoiding posterior collapse. The idea is to monitor statistics during training to detect potential posterior collapse, and to add in additional "calibration steps" for decoder output variance when signs of posterior collapse occur.

Empirically the proposed approach works better than KL annealing (a common approach to empirically alleviate posterior collapse) and the authors argued that the proposed approach is easier to tune.

---

> ### Author Response · Authors · 2025-01-20
> **Thank you!**
>
> Dear Action Editor,
>
> Thank you for your outstanding editorial service during the review and revision process and communicating this excellent outcome to us. We have uploaded the camera-ready version, incorporating all the points mentioned in your decision.
>
> Many thanks,
>
> The authors.